

# A decade of CH₄, CO and N₂O in situ measurements at Lauder, New Zealand: assessing the long-term performance of a Fourier transform infra-red trace gas and isotope analyser.

Dan Smale[1], Vanessa Sherlock[1], David W. T. Griffith[2], Rowena Moss[1], Gordon Brailsford[1],

Sylvia Nichol[1] and Michael Kotkamp[1]

[1]National Institute of Water and Atmospheric Research Ltd (NIWA), New Zealand

[2]Centre for Atmospheric Chemistry, University of Wollongong, Australia

*Correspondence to*: Dan Smale (dan.smale@niwa.co.nz)

**Abstract.** We present a ten-year (Jan 2007- Dec 2016) time series of continuous in situ measurements of methane ($CH_4$), carbon monoxide (CO) and nitrous oxide ($N_2O$) made by an in situ Fourier transform infra-red trace gas and isotope analyser (FTIR) operated at Lauder, New Zealand (45.04S, 169.68E, 370m AMSL). Being the longest continuous deployed operational FTIR system of this type, we are in an ideal position to perform a practical evaluation of multi-year performance of the analyser. The operational methodology, measurement precision, reproducibility, accuracy and instrument reliability are reported.

We find the FTIR has a measurement repeatability of the order of 0.37 ppb (1-sigma standard deviation) for $CH_4$, 0.31 ppb for CO and 0.12 ppb for $N_2O$. Regular target cylinder measurements provide a reproducibility estimate of 1.19 ppb for $CH_4$, 0.74 ppb for CO and 0.27 ppb for $N_2O$. FTIR measurements are compared to co-located ambient air flask samples acquired at Lauder since May 2009, which allows a long-term assessment of the FTIR data set across annual and seasonal composition changes. Comparing FTIR and co-located flask measurements show that the bias (FTIR minus flask) for $CH_4$ of -1.02 ppb ± 2.61 and CO of -0.43 ppb ± 1.60 are within the Global Atmospheric Watch (GAW) recommended compatibility goals of 2 ppb. The $N_2O$ FTIR flask bias of -0.01 ppb ± 0.77 is within the GAW recommended compatibility goals of 0.1 ppb should be viewed as a serendipitous result due to the large standard deviation along with known systematic differences in the measurement sets. Uncertainty budgets for each gas are also constructed based upon instrument precision, reproducibility and accuracy. In the case of $CH_4$, systematic uncertainty dominates whilst for CO and $N_2O$ it is comparable to the random uncertainty component.

The long-term instrument stability, precision estimates and flask comparison results indicate the FTIR $CH_4$ and CO time series meet the GAW compatibility recommendations across multiple years of operation, (and instrument changes), and is sufficient to capture annual trends and seasonal cycles observed at Lauder. The differences between FTIR and flask $N_2O$ measurements need to be reconciled. Trend analysis of the ten-year time series captures seasonal cycles, the secular upward trend of $CH_4$ and $N_2O$. The $CH_4$ and CO time series have the required precision and accuracy at a high enough temporal resolution to be used in inversion models in a data sparse region of the world.


## 1 Introduction

With the ubiquitous upward trend in anthropogenic greenhouse gas emissions (Stocker et al., 2013) there is increasing environmental and political impetus to respond. Under Annex 1 of the United Nations Framework Convention on Climate Change participating governments are required to report annual greenhouse gas emission

inventories. There is an increasing need to verify this bottom-up emission inventory approach with top-down approaches (Weiss and Prinn, 2016; Leip et al., 2018). A top-down approach is achieved by combining atmospheric greenhouse gas concentration (mole fraction) measurements and numerical atmospheric transport modeling so that surface flux estimates can be inferred. Such top-down approaches have already been undertaken to quantify national surface flux inventories of the main greenhouse gases ($CO_2$, $CH_4$, and $N_2O$) via

national surface in situ networks as in Peters et al. (2007), Ganesan et al. (2015) and Henne et al. (2016), or pan-national inventories using international greenhouse gas monitoring network databases (e.g. Cressot et al., 2016; Bergamaschi et al., 2015; Bergamaschi et al. 2018; and Pison et al., 2018). There is also a need for increased coverage in the southern hemisphere (Thompson et al.,2014; Wells et al., 2015), which is relatively data sparse compared to the northern hemisphere.

The National Institute of Water and Atmosphere (NIWA) Lauder atmospheric research station was established in 1961 for photometric observations of aurora airglow emission. The site was selected due to its relatively cloudless skies unaffected by light pollution and lack of air pollution. Such conditions also make it an ideal site for clean air trace gas observations. Atmospheric trace gas timeseries measurements started in the 1980s. The

current research focuses are on greenhouse gases, ozone depletion and UV/visible radiation. Lauder is a founding station in the Network for the Detection of Atmospheric Composition Change (NDACC), Total Carbon Column Observing Network (TCCON) and GCOS (Global Climate Observing System) Reference Upper Air Network (GRUAN) networks. It is also part of the Baseline Surface Radiation Network (BSRN) and is the primary New Zealand GAW station (GAW site ID: LAU).

The original reason for the establishment of greenhouse gas in situ measurements at Lauder were twofold. First, with the establishment of a TCCON site at Lauder in 2004 (Wunch et al., 2011; Pollard et al., 2017) it was an initial requirement that sites have co-located high precision continuous surface in situ measurements of $CO_2$ and $CH_4$. This was to provide a priori surface concentration constraints for the TCCON total column dry mole

fraction retrievals and provide an independent estimate of boundary layer $CO_2$ and $CH_4$. Second, it was to provide a complementary in situ measurement site to that at Baring Head, New Zealand (41.41 S, 174.87 E, 85m AMSL) (Brailsford et al., 2012) as a first step in a New Zealand carbon monitoring network. Measurements from these two sites have been used in a regional atmospheric inversion method determining $CO_2$ sinks and sources across New Zealand (Steinkamp et al., 2017).

A continuous in situ sampling system based upon closed cell Fourier transform mid infra-red spectroscopy (Griffiths and de Haseth, 2007) was chosen. The system selected was designed and built at the University of Wollongong (UoW) (Griffith et al., 2012; called G12 from now on). The FTIR can measure $CO_2$ (including isotopologues $^{12}C$-$CO_2$ $^{13}C$-$CO_2$ and $^{18}O$-$CO_2$ independently), $CH_4$, $N_2O$ and CO dry mole fractions

simultaneously with precision approaching and/or exceeding the GAW recommended compatibility goals





(GAW,2016). Measurements of $N_2O$, CO and $^{13}$C-$CO_2$, in addition to $CO_2$ and $CH_4$, have several benefits as $^{13}$C-$CO_2$ and CO provide additional information concerning carbon cycle source and sink attribution (van der Velde et al., 2018; Oney et al., 2018). $N_2O$ measurements in conjunction with $CO_2$ measurements allow estimation of surface $N_2O$ flux emissions (Kelliher et al. 2002; Laubach et al., 2016) which is pertinent given New Zealand's greenhouse gas emissions profile (MfE, 2017).

The Lauder FTIR was one of the first generation of such UoW FTIR systems using the Bruker IRcube FTIR spectrometer (Bruker Optics, Germany). The Lauder FTIR is of the same vintage as those deployed at the Darwin TCCON site (Deutscher et al2010a), Cape Grim (Griffith et al., 2011), University of Wollongong (Buchholz et al., 2016) and similar to the system operated at the University of Heidelberg Institut fur Umweltphysik (IUP, Hammer et al., 2013a; called H13 from now on). In 2013 the UoW FTIR system was commercialized, in a joint venture between UoW and Ecotech (Australia) and marketed under the name Spectronus.

Previous work has characterized performance and data quality of the UoW FTIR systems on time scales ranging from short lived field campaigns and up to 4 years. H13 provided an extensive performance evaluation of the IUP FTIR in laboratory and campaign-based studies over a period of 8 months. Comparison of the FTIR performance to other in situ instrumentation has also been conducted in Griffith et al. (2011), Hammer et al. (2013b), Vardag et al. (2014), and Lebegue et al. (2016). CO performance has only been evaluated by Griffith et al. (2011), with inconclusive results due to variable CO amounts in the calibration tank resulting in poor accuracy. In all these studies $CH_4$ mean differences were within the GAW compatibility recommendations, whereas for $CO_2$ and $N_2O$ differences were overall marginally higher than the recommendations. Only in Lebegue et al. (2016) was the FTIR operated for longer than a year. The Lauder FTIR was part of a GAW quality assurance strategy performance audit using travelling standards (Zellweger et al., 2016) in which it was the only FTIR. The audit results show the FTIR $CO_2$ and $CH_4$ measurements to be comparable to other measurement types ($N_2O$ and CO were not assessed). Other studies have shown the durability and reliability of the FTIR during field campaigns (Deutscher et al., 2010b; Laubach et al. 2016; Sonderfeld et al., 2017).

Despite this promising work, questions remain concerning FTIR performance and stability over longer time periods, such as multiple years to decades. Proven reliability over such periods is required if the FTIR is to be deployed as part of long term monitoring networks. Studies by Buchholz et al. (2016) and Té et al. (2016) both use data from the two FTIRs operated at UoW with a combined duration of 3.3 years. The longest continuous FTIR temporal dataset published to date is 4 years in length (Vardag et al.,2016).

In this study we investigate the Lauder FTIR $CH_4$, CO and $N_2O$ precision, repeatability, and accuracy over 10 years of operation. We also comment on the reliability of the FTIR, looking at more day-to-day operational issues than previous studies, such as regular maintenance, instrument failures and areas for potential improvement.





The FTIR measurements are then compared to co-located flask air sample measurements, which were initiated to provide a cost effective independent data set. Simplistic time series analysis is performed to provide an estimate of the annual trend and seasonal cycles and to ascertain if the FTIR can observe such atmospheric change on such time scales. Investigation into the FTIR $CO_2$ and $\delta^{13}C$-$CO_2$ measurement performance along with

comparisons to co-located independent $CO_2$ (Steinkamp et al., 2017) and $\delta^{13}C$-$CO_2$ measurements will be reported in a separate study.

In sections 2 and 3 we describe the Lauder atmospheric research station and the in situ instrumentation at the site. Section 4 details the air inlet sampling system, common to all in situ sampling instruments. In Sect. 5 we

introduce the FTIR, describe significant upgrades to the instrument and issues associated with its operation. We assess the long-term stability of precision and accuracy, along with calibration methods. Uncertainty budgets are also constructed. In Sect. 6 we detail the Lauder flask sampling program and then compare the FTIR to flask measurements in Sect. 7. In Sect. 8 we perform a simple trend analysis on calibrated FTIR air sample data taken in so called 'baseline' conditions and deduce annual trends and seasonal cycle for each species. In Sect. 9 we

offer a concise summary of the work undertaken.

## 2. Site location

The Lauder atmospheric research station (45.038° S, 169.684° E, 370 MASL) is in Central Otago, South Island of New Zealand (see Fig. 1). A description of the geography of the site and surrounding region is given in Steinkamp et al. (2017) and Pollard et al. (2017). The station is located in a broad valley surrounded by pastural

farmland with low stock density, with no nearby industrial emission sources. Clear skies, low viewing horizon geometry and lack of air pollution were the original reasons for the site selection. The nearest town, Alexandra, is 35 km to the south and has a population of approximately 5300. The climate is considered semi-arid and continental. Westerly winds dominate the wind flow over the South Island of New Zealand. At Lauder, predominant moderate breezes (greater than 5 ms$^{-1}$) are from the west, whilst nocturnal light breezes are mainly

from the north-east, down valley. Lauder air history maps calculated from back trajectory analysis (Steinkamp, et al. 2017) show that much of the sampled air originates (since last boundary layer contact) from the West coast of the South Island, a heavily native forested region. All these conditions make Lauder an ideal site to take baseline measurements (baseline conditions are defined Appendix D).

## 3. Instrumentation

In situ ground level greenhouse gas measurements started at Lauder in August 2006 with the installation and commissioning of the FTIR. This was followed by installation of a NDIR Licor-7000 instrument in June 2008 to provide $CO_2$ comparison measurements (Steinkamp et al., 2017). A flask sampling system was added in May 2009. Flask air sample analysis provides cost-effective independent measurements of $CH_4$, CO and $N_2O$, and

additionally provides another independent in situ $CO_2$ and $\delta^{13}C$-$CO_2$ data set. It is planned that all three measurement systems continue to operate in parallel into the foreseeable future. A description of the air inlet system, FTIR and the flask sampling system will be given in the next three sections. A set of meteorological



sensors were added to the in situ sampling mast in September 2008 to provide wind, temperature and humidity measurements at different heights. Prior to this, meteorological data from the Lauder NIWA climate station was used, located 90 metres from the mast. The in situ sampling instruments are housed in a temperature controlled building (see Fig. 1b). The 10-metre-high in situ sampling mast is located 33 metres to the north of the building

to minimise the impact of southerly wind flow disturbance.

## 4. Air inlet sampling system

A detailed description of the current air inlet system and meteorological sensors can be found in Appendix A of Steinkamp et al., (2017). The original air sampling system consisted of 60 metres of 3/8 inch copper tubing, 30 metres of which was underground. The inlet on top of the sampling mast was connected directly to the FTIR. A

moisture trap was located at the base of the sampling mast. With the installation of the Licor-7000 in June 2008, a 4-port manifold and roughing pump were added, thus both instruments use a common sampling line. With this system air is drawn from the 10-metre inlet height at a rate of 10-15 $Lmin^{-1}$. Residence time is approx. 35 seconds. Manifold pressure is typically 40 hPa below atmospheric pressure. Manifold pressure is monitored with an analogue mechanical vacuum gauge. Four ¼ inch stainless steel (SS) tubes are welded perpendicular to the

main body of the manifold providing connection points for sampling systems, each with a terminating ball joint valve. Swagelok components and joins are used through-out. Short lengths of polytetrafluoroethylene (PTFE, aka Teflon) tubing are used to connect instruments to the manifold to electrically isolate them from the mast to minimise potential lightning strike damage. The copper tubing was replaced with 3/8 in SS tubing in November 2012. This tubing is all above ground. It should be noted that the air inlet delivery system does not dry the air,

this is done on an instrument by instrument basis. A flask sampling system was installed in May 2009. With all three in situ instruments connected to the manifold the total maximum draw is 8.1 $Lmin^{-1}$ (3.5 $Lmin^{-1}$ FTIR, 2.6 $Lmin^{-1}$ Licor-7000 and 2.0 $Lmin^{-1}$ flask sampling). This combined instrument draw is less than the manifold flow. Instrument cross sampling is not a concern.

The air sampling line is checked for leaks every 6 months, taking approx. 2 hours. During this time measurements are suspended. The line is visually inspected then capped and pressurised with dry nitrogen to 300 psig to help locate any leaks. The moisture trap at the base of the mast is emptied (approx. 1-5 mL). The mast is lowered, and the inlet coarse filter cleaned. The meteorological sensors are also attended to. The front-end and sample line roughing pumps are tested weekly and pump diaphragms visually inspected every 6 months. A torn

diaphragm is the most common cause of failure.

## 5. FTIR

In this section we outline of how the FTIR works, routine operation, calibration procedures and detail instrument upgrades over time. Long term FTIR performance i.e. reliability, accuracy, precision and repeatability is evaluated.

The Lauder FTIR is based on FTIR systems described in Griffith et al. (2011), G12 and H13. A complete description can be found in these references. It was the second FTIR built at the University of Wollongong



(UoW) chemistry department using the Bruker IRcube, with a thermoelectrically cooled mercury cadmium telluride (HgCdTe) detector. These components supplanted a previous FTIR system based upon a Bomem MB100 (ABB Bomem, Canada) interferometer and a HgCdTe detector cooled with liquid nitrogen (Esler et al., 2000). These changes made the FTIR more reliable, with less operator intervention, and with greater

measurement precision. Many significant changes to hardware, data acquisition and spectral processing have happened during instrument deployment at Lauder, and these are described in detail in the following sections.

**5.1 Hardware**

The FTIR analyser was originally installed in late August 2006 followed by a 4-month commissioning phase in which acceptance testing was performed along with training in instrument operation and data analysis.

Continuous air sample measurements started in January 2007. Since installation, the FTIR has undergone several improvements in both hardware and software since that time. We first describe the original configuration and those components which have not changed, then incremental improvements over the 10 years of operation (Jan 2007- Dec 2016).

The unchanging core of the FTIR analyser consists of a Bruker IRcube interferometer (CaF$_2$ beam splitter, resolution 1.0 cm$^{-1}$) coupled to a 3.5 L glass multi-pass White cell (PA-24, Infrared Analysis, USA). The IRcube has an internal globar, mid infrared radiation from which passes through the cell traversing an optical path of 24 metres. A thermoelectrically cooled HgCdTe detector (Teledyne Judson Technologies, USA) measures mid-IR radiation over the wavenumber range 1750-6750 cm$^{-1}$. Interferogram acquisition and spectrum calculation is

performed through Bruker's proprietary acquisition software, OPUS, and the analyser's data acquisition software (described below). The IRcube and cell transfer optics is continually purged with dry nitrogen (100 mL min$^{-1}$) to displace the relatively humid room air and prevent build-up of CO in the optical path outside the cell.

The FTIR enclosure is thermostatically controlled, with a manual set point at 34.0°C. Cell temperature was

originally monitored with a LM335 integrated circuit sensor attached to the outside of the cell (resolution 0.1 °C), later replaced with an in-cell RTD sensor as described further below. The cell pressure is measured with a piezo transducer (model series 902, MKS Instruments, USA, resolution 0.13 hPa). The measured cell temperature and pressure are used in quantitative spectral analysis, and in the subsequent conversion of the retrieved concentrations to mole fractions.

A schematic of the initial FTIR gas handling system is presented in Fig. 2a. The gas handling system delivers gas to the cell from one of four software-selectable inlet valves, two of which were passed through a drying system as described below, and two of which were undried by the analyser. Originally, there were two air inputs, the air sample line and a working standard (WS) which is used as part of the calibration procedure. A target

cylinder (TC) was later added to provide a means to monitor FTIR reproducibility and accuracy. Air samples passed through the drying system whilst WS and TC tank air remained undried by the analyser (both WS and TC are dried at the point of collection). Dual stage scientific regulators (model 1-SS30-590-D4T, Scott Marrin Inc., USA) provide a step down from the cylinder pressure of 2000 psig to a stable low side pressure in the range of 5-20 psig.





Electronically actuated solenoid valves (Models 6013 & 6014, Burkert, Germany) controlled by the FTIR data acquisition software allow manipulation of gas flow and delivery. Air samples are dried using a 24-inch Nafion dryer (model MD-070, Permapure, USA) in series with the chemical desiccant anhydrous magnesium

perchlorate ($Mg(ClO_4)_2$). The backflush for the Nafion dryer was provided by the (dried) sample air exiting from the measurement cell at reduced pressure. Air samples are dehydrated to less than 20 ppm. Cylinder gases were not dried. All gases pass through a 7 µm particulate filter prior to reaching the White cell. A vacuum pump (model MV2NT, Vacuubrand, Germany) at the exit to the cell and Nafion backflush provides the required pressure gradient to allow gas flow, to evacuate the cell for spectrum background measurement and to provide

the Nafion dryer backflush.

Measurements are taken in two modes of operation: static mode and flow mode. In static mode, the cell is evacuated then filled with gas to a defined pressure. The cell is then closed, and spectral measurements are made. In flow mode, gas is continually drawn through the cell at a set flow rate whilst spectral measurements are made.

In the initial instrument configuration, in flow mode the flow rate was controlled by a manual set needle valve located downstream of the cell and the flow rate was monitored by a mass flow meter (model 820 series, Sierra instruments, USA). In flow mode the cell pressure and flow are not independent. Reducing the flow increased cell pressure and vice versa. The cell pressure was also proportional to the input delivery pressure. In addition, the magnesium perchlorate solidifies over time as the desiccant dehumidifies gas reducing both flow and

pressure in the cell. There is a slow constant change in cell pressure and flow. Due to the air sampling configuration at Lauder sample air is measured in flow mode. Cell pressure is in direct proportion to the inlet manifold pressure which in turn is proportional to atmospheric pressure. Cylinder gas measurements are conducted in static mode to reduce gas consumption. The static mode cell pressure set point is altered at regular intervals to be similar to cell pressure during sample air measurements. This is done to reduce residual pressure

sensitivity (RPS) (detailed in Sect. 5.7.1).

The data acquisition system is the same as that described in G12. The entire analyser is controlled by the custom coded software ('Oscar', V9.1.8) developed at the UoW. Oscar is written in Visual Basic 6. It schedules the measurements, gas input selection, operates the gas handling valves, logs instrument parameters (pressure,

temperature, and flow) and interacts with OPUS. Oscar also actuates the spectral retrieval analysis software to perform real time processing after each measurement. Details about the spectral retrieval software are given in Sect. 5.6.

### 5.2 Significant instrument changes

There have been continual improvements to the FTIR and air inlet systems over the working lifetime of the

instrument at Lauder, some of which have been incorporated into the current commercial design. The upgrades have all lead to an improvement in cell temperature and pressure stability. The main improvements were replacing the external cell temperature sensor with a high-resolution sensor located inside the cell, independent control of cell pressure and flow rate, rerouting of internal tubing so that cylinder gas and air samples are all treated equally and dried, and lastly, a front-end pump to deliver sample air at a constant pressure.



### 5.2.1 Monitoring cell temperature

Cell temperature was originally monitored using a sensor based upon a generic LM335 integrated circuit attached to the outside of the cell. It had a resolution of 0.1 °C. This approach assumes that the external cell wall temperature is the same as the gas in the cell, and the cell wall is unaffected by the temperature of the FTIR enclosure. Alone, the coarse resolution of the LM335 introduces a non-insignificant uncertainty in the retrieved $N_2O$ dry mole fraction of approx. 0.1 ppb at 320 ppb (at typical cell pressure and temperature) but less significant for CO (approx. 0.02 ppb at 60 ppb) and $CH_4$ (approx. 0.6 ppb at 1800 ppb). In September 2010, a PT100 resistance thermometer detector (RTD) was inserted into the cell to measure gas temperature invitro. The PT100 is coupled to a PR4114 universal transmitter (PR Electronics, Denmark) providing a temperature resolution of 0.002 °C. This allows a more precise and responsive direct measurement of the gas temperature in the cell.

Figure 3 clearly shows a change in recorded cell temperature when the sensors were swapped in September 2010. There is a significant bias (approx. 1.3 °C) between the two temperature measurements. This is not of concern as the bias is systematic and compensated for during the calibration process. The 1-sigma standard deviation (1σ) in the PT100 is 0.05 °C compared to 0.3 °C for the LM335. The PT100 is more stable and less susceptible to changes to FTIR enclosure temperature fluctuations and more indicative of cell gas temperature. As part of the April 2013 upgrade the invitro PT100 was replaced with a Type-J thermocouple. Even though the thermocouple has a faster response time, no significant changes in temperature precision were seen.

### 5.2.2 Independent control of cell pressure and flow rate

In the initial instrument configuration, in flow mode cell pressure and gas rate flow are coupled so that adjusting one affects the other. Control of either was by manual adjustment of the needle valve located downstream of the cell (Fig. 2a). The cell pressure during sample air measurements is dependent on the air inlet system manifold pressure which in turn is proportional to atmospheric pressure. As the desiccant solidified it also caused a reduction in both cell pressure and flow. Continual adjustment was required to keep both cell pressure and flow within a given range. More importantly, since the WS is measured in static mode, and the cell filled to a defined pressure, there was always a difference between sample air and calibration gas pressures. Differences up to 50 hPa were common.

The solution to decoupling cell pressure and flow and providing cell pressure stability was to replace the needle valve and mass flow meter with two mass flow controllers (MFC, Model 3660, Kofloc, Japan). One MFC was installed upstream of the cell and the other downstream, as shown in Fig. 2b. The upstream MFC controls the flow rate through the cell, whilst the downstream MFC is constantly adjusted via a Proportional-Integral-Differential control loop to maintain constant cell pressure. The upgrades also correct for the reduction in flow and pressure due to the desiccant solidifying. Cell pressure and flow rate can be set independent of each other. The upgrade was done in April 2013. The effect of this change is seen in Fig. 4a. Prior to the upgrade the standard deviation in cell pressure and flow were 36 hPa and 0.03 Lmin⁻¹ respectively. After the upgrade cell pressure and flow standard deviations were 0.001 hPa and 0.005 Lmin⁻¹. There is also a significant reduction in





sample air cell pressure and calibration gas cell pressure bias (Fig. 4b). The bias reduces to 0.02 hPa, resulting in a negligible pressure residual cross sensitivity correction (see Sect. 5.7 for more details).

### 5.2.3 Inlet port reconfiguration

During the April 2013 upgrade the inlet port lines were reconfigured so that all four inlet lines are equivalent and pass through the internal drying system (Fig. 2b). This allows cylinder gas to be dehydrated to a level equal to that of the air samples. Prior to this change, it was assumed cylinder gas was pre-dried, or an external drying system was required. Differences in water content can introduce measurement bias, such as encountered in Zellweger et al. (2010).

A 16-port multi-position valve (model EMTSD16MWE, Valco, USA) was added in June 2015. This allows multiple cylinders to be attached to a single analyser inlet. Whilst not directly affecting the FTIR performance, it takes less time to measure multiple cylinders without detaching any tanks. The FTIR data acquisition software has modules to control Valco multi-valve operation.

### 5.2.4 Addition of a front-end pump to provide a stable inlet pressure

A FTIR front-end pump was added in September 2013. It is placed between the air sampling inlet manifold and the FTIR inlet ports (Fig. 2b). The purpose of the front-end pump is two-fold, to provide additional sample delivery pressure stability and to increase cell pressure above that deliverable by the air inlet sampling system. The front-end pump consists of a diaphragm pump (model N86KNE, KNF Neuberger, Germany) followed by an
atmospheric bypass dump. The bypass has a needle valve and poppet check valve (model SS4CA-3, Swagelok, USA) attached in series. The front-end pump and the FTIR downstream MFC together effectively act as an inlet pressure controller. The needle valve provides a coarse inlet pressure setting which is then finessed to the cell pressure set point via FTIR MFC control. The poppet check valve is a safety precaution to avoid over pressurising the cell. The front-end pump draws sample air from the manifold at 3.5 Lmin$^{-1}$, most of the sample
air exits via the bypass, while a constant input pressure is provided to the inlet (approx. 1200-1300 hPa). Such inlet pressure control was also employed by H13 using an electronic pressure controller. With the new front-end system, cell pressure is set to 1100 hPa for air sample measurements (standard operation conditions will be described in the next section). Operating with a cell pressure above atmospheric pressure provides positive pressure making leak detection easier and minimises the effects of any leaks. The signal to noise ratio (SNR)
also increases due to increased absorption. The front-end pump pressure setting is monitored (and set if necessary) every 2-3 months during as part of the regular maintenance schedule.

### 5.3 Standard operating conditions

Overall, routine operation of the FTIR has remained very much unchanged since measurements started. Whilst upgrades have contributed to changes in operating procedures, the underlying instrument set up has been stable.
The FTIR is configured to continuously measure air samples interspersed with regular cylinder measurements for calibration and quality assurance. This is one of the simplest FTIR configurations the FTIR can be deployed in (other deployment configurations are described in G12).




Air sample measurements are taken in flow mode. Air is drawn into the White cell at 0.5 Lmin⁻¹ at the defined pressure (originally 870 hPa, then 1100 hPa after the April 2013 upgrades). At a rate of 0.5 Lmin⁻¹ and with the White cell volume of 3.5 L, the e-folding time (Winderlich et al., 2010) is approx. 7 minutes, meaning sequential

flow mode sample measurements (10-minute averages) are not completely independent of each other. FTIR temperature is stabilised at 34.0 °C ± 0.2. The heater unit has a duty cycle of approx. 40%.

The spectra acquisition settings have remained unaltered over the entire period. Spectra acquisition consists of 721 coadded scans averaged over 9.5 minutes. All spectra are taken with a resolution of 1.0 cm⁻¹ and with an

aperture of 1.5 mm. The effective field of view is 21.7 mrad (full angle). The Happ-Genzel apodization function is applied to the collected interferogram with a Mertz phase correction. The spectra also exhibit minor etalon channelling of approx. 0.005% signal strength with a period of approx. 5 cm⁻¹. The channelling is stable in both period and amplitude and is inconsequential, but a noted feature that should be diagnosed in each FTIR. The resulting spectra have an SNR of the order of 15000-20000.

Real time quantitative spectral analysis occurs after each spectrum collection (details in Sect. 5.6). This takes approx. 30 seconds, giving an overall collection and processing time of just under 10 minutes resulting in 144 measurements per day (if no calibrations are performed). Scheduling is organised into 30-minute cycles, hence three 10-minute sample measurements per cycle. Each spectrum is saved with a unique filename and the results

of the spectral analysis are added to a daily summary file. The results are also displayed in real time (updated every 10 minutes). Whilst the displayed results of the spectral analysis are not calibrated they are an extremely useful diagnostic.

Up until February 2014 calibrations were performed daily. The calibration procedure consists of two parts,

background spectrum collection followed by WS measurements. WS spectra acquisition parameters are identical to that used in sample air measurements. A background spectrum is measured after evacuating the cell to approx. 1 hPa, or until 180 seconds has passed, whichever is reached first. The background spectrum is then stripped of remnant water absorption features (explained in appendix B). During the 9.5-minute background spectrum acquisition sample air is continuously drawn through the FTIR system via bypass tubing. This flow keeps the

sample desiccated and at a stable temperature. The acquired background spectrum is subsequently used to produce both sample and calibration transmission spectra.

On completion of background spectrum measurement, the WS tank is measured in static mode. Static mode is used to reduce gas consumption as each cell fill uses approx. 3.5 L of gas. Prior to WS tank measurement the cell

is flushed with 200 hPa of WS gas then the cell is re-evacuated to 1 hPa and filled to the prescribed pressure set point. Filling takes approx. 60 seconds. A latency period of 60 seconds after filling allows the cell pressure and temperature to stabilise reducing the effects of thermodynamic disequilibrium (H13) after which spectra are acquired, saved, analysed and results written to a daily file. A single 10-minute WS spectrum is acquired and analysed. The resulting data are then used in post-processing calibration procedures. The entire calibration cycle

(background and WS measurements) takes approx. 25 minutes, fitting into the 30-minute cycle block.





Sample measurements resume after the calibration. The cell is evacuated, flushed with sample air then filled to the prescribed pressure set point and allowed to settle using the same procedure as in the WS measurements. Flow mode is then activated, and sample spectra are acquired. The first spectrum acquired after calibration is filtered out of the final processed dataset as the water content is greater than normal due to the cell still not reaching moisture equilibrium.

TC measurements are conducted in the same manner as WS measurements, except a background spectrum is not taken. When daily TC measurements started there was a reduction in sample collection time by another 30 minutes. Overall, in each 24-hour period 1.5 hours were used in calibrations activities. Calibrations were scheduled to be performed at 2am to avoid interfering with daytime sample collection. Under this calibration regime it took approx. 1.5 years before the WS and TC tanks reached a pressure of 500 psig. At 500 psig, the tanks are replaced.

The FTIR upgrade in April 2013 allowed significant changes in the calibration procedure. In February 2014, a new calibration procedure was constructed to allow flow mode calibration and TC measurements every week. The change from daily to weekly calibrations is within the recommendations of H13. Flow mode calibrations aligns the tank measurement procedure with that of air sample measurements. Background spectrum acquisition remains unaltered, after that the evacuated cell is then filled with tank gas to 1100mb over a period of 420 seconds. A latency period of 300 seconds follows. The combined slower fill rate and longer settling time allows cell temperature and pressure to stabilise with a significant reduction in thermodynamic disequilibrium. The effect of thermodynamic disequilibrium has minimal impact on $CH_4$, CO and $N_2O$ spectral analysis but significant for $CO_2$. Once the cell is filled, tank gas flows at a rate of 0.5 Lmin$^{-1}$ during which spectra measurements are taken. Four 10-minute spectra are collected. The first is not used, effectively allowing another 10 minutes for the FTIR to stabilise. The entire calibration process takes 1.5 hours using approx. 24 L of WS gas which is equivalent usage to a week of daily static mode calibration measurements. Also, collection of three sequential WS tank measurements (compared to the previous single static mode measurement) allows calibration reproducibility to be assessed. TC measurements are also conducted every week in flow mode. This takes an additional 1 hour making a total of 2.5 hours per week for calibration and quality assurance checks.

In this configuration the FTIR can operate autonomously for a week. User intervention is required each week to start the combined WS and TC calibration measurement schedule, then once completed to restart routine air sample line measurements. Extended periods of automation are possible (such as at remote unmanned sites) with a different measurement schedule but given that the FTIR is located on-site and accessible, regular checks and intervention are not an issue.

**5.4 Routine maintenance**

Like all instruments, regular maintenance is required and is essential for optimum performance. We have found that regular maintenance is minimal and have had only one component failure over the decade of operation. The



most common interruption to measurements is replenishing consumables. In this section we describe routine maintenance tasks.

### 5.4.1 Nitrogen purge

Dry nitrogen to purge the IRcube and cell transfer optics is used at a rate of approx. 100 mL min$^{-1}$. The dry nitrogen cylinders (2000 psig) last 6-8 weeks. Cylinder changeover takes less than 5 minutes and can be completed without the need to interrupt measurements. In the original configuration $N_2$ flow was controlled with a needle valve and monitored with a rotameter (model FR-2000, Brooks Instrument, USA), giving coarse flow control. An MFC (model 80SD-5, McMillan, USA) was then installed in February 2015 providing better flow

control and gas management.

### 5.4.2 Chemical desiccant replenishment

Symptoms of reduced moisture absorption by the desiccant is an increase in $H_2O$ in the cell and prior to the decoupling of cell flow and pressure, a reduction in both flow and pressure as the desiccant solidifies. In standard operating conditions $H_2O$ in the cell is less than 10 ppm (Fig. 5a). A rise of 5ppm (or greater) over the

course of a week is indication that the desiccant needs replacing. The chemical desiccant is replaced every 3 months. The initial desiccant cartridge consisted of Drierite (calcium sulphate impregnated with cobalt chloride, 60 g) and granular magnesium perchlorate (60 g) in series separated by glass wool. The upstream Drierite was to provide a visual indicator when to replenish the desiccant. We found that the $H_2O$ concentration from the spectral analysis was a considerable better indicator. Magnesium perchlorate is now the sole chemical desiccant

used and 60 grams is still sufficient. Doubling the amount of desiccant did not increase the cartridge lifetime as one of the limiting factors is the cartridge cross sectional area.

Changing the desiccant requires removal of the cartridge from the FTIR. The cartridge is attached to the FTIR via quick release fittings (model QC-4, Swagelok, USA). After cartridge replenishment and reinstallation

moisture levels of 50 ppm (or greater) are present due to inherent moisture in the replenished cartridge (due to being exposed to humid laboratory air) and associated tubing. It takes approx. 5 days for the cell to dry to less than 10 ppm (Fig. 5b). As the cell dries out we filter out data where calibration and sample measurements have a difference in $H_2O$ content greater than 10 ppm. This is a conservative approach with approx. 5 days of data every 3 months not used. As a technical aide, it should be noted that the downstream cartridge filter sinter

element accumulates powdered magnesium perchlorate which over time solidifies and reduces flow. The sinter element is cleaned each time the cartridge is replaced.

### 5.4.3 Pressure sensor calibration

Every 3 months (to coincide with chemical desiccant replenishment) the cell pressure sensor is tested, and if

needed it is recalibrated. We found this necessary as during the initial installation and commissioning period (August - December 2006) the sensor was 3.6 hPa too high. Whilst a pressure offset would be common to both calibration and sample measurements, and effectively cancelled out during the calibration process, an accurate



cell pressure reading is preferable. Routine checks did not start until mid-2012; up until then it was (wrongly) assumed that sensor calibration would hold, and only sporadic checks were performed. The routine pressure sensor checks show that sensor drift can be up to 2 hPa over a 2-month period, and as high as 4 hPa, over a 3-year period (Fig. 6). We do not know the cause of the drift.

Both the pressure sensor span and offset are checked using independent pressure sensors. To check the FTIR pressure sensor offset a capacitance manometer (model 222BA Baratron®, MKS instruments, USA) is connected to the exit port of the cell, then the cell is evacuated to < 1 hPa. The FTIR pressure sensor offset is adjusted to get agreement.  To check the span, the cell is then filled and allowed to equalise at atmospheric

pressure. The cell pressure is then compared to an external independent pressure sensor (model PTB110, Vaisala, Finland) located next to the FTIR. The FTIR pressure sensor span is adjusted to get agreement. Both external pressure sensors have traceability records to the NIWA metrology standards. For the majority of comparisons, the offset was the only adjustment required. We recommend that FTIR systems using the MKS Series 902 pressure sensor are checked regularly.

### 5.4.4 IRcube metrology laser replacement and internal globar.

The IRcube has an internal single mode 0.84 mW 633 nm helium–neon (HeNe) laser to provide an accurate measurement of scanner arm displacement crucial for interferogram acquisition. The HeNe laser is classified as a consumable as it has a finite lifetime hence replacement is regarded as a routine but infrequent maintenance

issue. There have been three laser replacements over the 10-year period due to laser failure, in October 2009, April 2011 and July 2013. In all three cases, the replacement laser was a Melles Griot 05-LHP-211 but other compatible products could be used (e.g. Lasos LGK-7604P, Lumentum 1107P). A proactive approach is possible by replacing the laser before it fails. This is the recommended approach but must be balanced with the incurred extra expense.

Installation of a new laser is straight forward. Laser pointing is adjusted to maximise laser signal strength whilst also maximising interferogram signal strength and minimising interferogram asymmetry. Diagnostic tools for laser alignment are provided as part of the OPUS software. Installation and alignment takes less than 3 hours.

The internal MIR globar (12V, 20W) has a designated factory life time of approx. 5 years (44,000 hours). After 10 years of continual operation the globar has not been replaced. The reduction in signal level (Fig. 7a) could be a sign of a diminishing globar output, but as we see no degradation in spectra SNR we have decided not to replace it.

### 5.5 Interferometer performance

There has been no published long-term performance evaluation of the Bruker IRcube as part of the FTIR system. Assessing the quality of the acquired interferograms and associated spectra assists in diagnosis of instrument issues. Changes in spectral SNR and/or instrument line shape (ILS) degradation will propagate through to



spectral analysis, hence retrieved cell gas dry mole fractions. Since changes in the IRcube will equally affect both sample and calibration spectra acquisition, the calibration procedure will mitigate such effects, but will also mask them, thus only by looking at the raw data will we be able to assess instrument spectral acquisition performance.

For such diagnostic purposes, WS tank interferogram and spectra signal levels, SNR and ILS parameters are calculated. These are displayed in Fig. 7. Over the 10 years of operation, the interferogram zero path difference (ZPD) intensity has been dropping, interspersed with periods of stepwise gains. The stepwise gains are associated with laser replacements and resetting of the ZPD reference position. The cause of the continual

decline in ZPD intensity is unknown, but to speculate, it could be degradation in the mid infra-red (MIR) globar intensity, internal optic transmittance or $CaF_2$ beam splitter transmittance. And as expected the associated interferogram spectrum mean signal level is also declining but does not have piecewise steps as the spectrum is normalised in the Fourier transform. Both SNR and spectrum signal level vary slowly indicating good reproducibility and stability. Considering that the mean signal level decreased over time it is interesting that the

SNR increased indicating that the reduction in noise was greater than signal degradation with cause unknown.

The field of view (FOV) and spectrum phase are fitted to monitor of linewidth and asymmetry. The fitted FOV and phase are displayed in Fig. 7c. There is a gradual decline in phase, but the overall phase is very small ($< 0.01$ rad) indicating a stable near symmetric ILS. The small step changes in phase are related to a change in the cell

temperature sensor, laser replacement and operation of the FTIR with a different FOV. The theoretical FOV of the IRcube is unvarying at 21.73 mrad, (apart from brief testing period in mid-2011), thus any deviations in the fitted FOV indicate an issue in spectra acquisition or analysis. Prior to September 2011 the calculated FOV was lower than expected but still stable. This was because the background spectra acquisition aperture setting (3 mm) differed from the sample spectra acquisition aperture setting (1.5 mm). The background aperture size was set to

1.5 mm in September 2011. After this change the fitted FOV agrees well with the physical FOV.

A decade of IRcube diagnostics illustrates the stability of the interferometer. To date, replacing the internal metrology laser (detailed in Sect. 5.4.4) is the only regular maintenance required. The IRcube failed once due to a burnt-out resistor in the 24 VDC detector power supply rail. Apart from this, no other components have needed

replacing.

### 5.6 Quantitative spectral analysis

Only a summary of the FTIR quantitative spectral analysis method is given as a succinct introduction is provided in G12 with detailed descriptions in Griffith (1996) and Griffith et al. (2003). Details specifically related to the

Lauder FTIR spectral analysis will be covered.

Cell gas column concentrations (mol m$^{-3}$) are calculated from the spectra by iterative non-linear least squares fitting of the measured spectrum with that of forward modelled theoretical spectrum. The code used to perform this analysis is called MALT (Multiple Atmospheric Layer Transmission) (Griffith, 1996). Input parameters to





MALT include the instrument line shape function (ILS), cell optical path length, cell pressure and temperature, an a priori estimate of gas mole fractions and absorption line parameters sourced from the HITRAN 2004 database (Rothman et al., 2005). On a spectrum by spectrum basis, all inputs and a priori values to MALT remain constant except for cell pressure and temperature (which are specified, not fitted). Broad spectral regions

of 100-200 cm$^{-1}$ are analysed. The selected spectral analysis regions are optimised per species. The retrieval strategies used at Lauder are the same as in G12. The retrieved cell gas species concentrations are converted to mole fractions using the ideal gas law (G12 Eq 1.), then to dry air mole faction ($\chi_{dry}$) using Eq 2. in G12. All subsequent analysis is conducted using $\chi_{dry}$ unless otherwise stated.

Successive versions of MALT (from V5.3 to the current V5.5) have been used as part of the Lauder FTIR system. MALT input files are edited to match the Lauder FTIR physical parameters (i.e. field of view, spectral resolution, cell optical path length). There have been two main changes to the retrieval strategy: 1) a reduction in CO and N$_2$O residual cross sensitivity to $^{12}$CO$_2$ by fitting CO and N$_2$O in a different spectral region. This also has the fortuitous effect of reducing N$_2$O nonlinear cross sensitivity to cell pressure, and 2) improved spectral

fitting of water vapour in background spectra. Details of these two changes are found in Appendix A and Appendix B respectively.

**5.7 Residual cross sensitivities**

As detailed in G12 and H13 the calculated raw (pre-calibrated) species dry mole fractions have a small non-

trivial residual dependence on the input parameters used in the quantitative spectral analysis. These empirical residual cross sensitivities (RCS) are attributed to imperfections in the measured spectra, systematic uncertainties in the spectroscopic database, the spectral analysis procedure and uncertainties (systematic and random) in temperature and pressure measurements.

For each species the RCS for each parameter i.e. cell pressure, cell temperature, cell flow and water vapour, as well as species cross-sensitivity, need to be experimentally derived. From these experiments a simple linear regression is sufficient to parametrise the RCS (H13). The calculated RCS is then used to calculate a correction to be applied to the measured dry mole fraction, as in Eq. (1). Where $\chi_{rcs\_corr}$ = corrected dry air mole fraction, $\chi_{dry}$ = raw spectra dry mole fraction, RCS$_z$ =residual cross sensitivity term between $\chi_{dry}$ and parameter Z. Z$_o$ =

reference parameter amount. In our application, we use the most recent calibration parameters as Z$_o$, thus all corrections are relative to the conditions calibrations were taken in.

$$\chi_{rcs\_corr} = \chi_{dry} - \sum(RCS_z * (Z - Z_o)), \qquad\qquad (1)$$

Results from extensive tests by H13 (Table 1 in H13) give an indication of expected RCS values. Caution should be taken as such RCS values are not generic and should only be applied to FTIR systems of the same model and analysis software. This is because RCSs differ depending on sensor placement (H13), cell wall effects introducing water vapour hysteresis and a dependence on the spectroscopic database used. The Lauder FTIR has sufficient differences to that used by H13 to warrant the need for experimental derivation of RCSs. In all



instances it is best to minimise RCS corrections by making sure standard operating conditions are as stable and similar as possible across both sample and calibration measurements.

For the Lauder FTIR, only cell pressure RCS is used. All water and cell temperature RCS experiments were
inconclusive due to the demanding nature of the tests which could not be resolved. In both cases uncertainty in the results were too large, the main issues being time lag and water vapour hysteresis. Given inconclusive results we decided to omit temperature and $H_2O$ RCS corrections. This is not uncommon, both H13 and Lebegue et al. (2013) also found such experiments challenging. With strict QC/QA filtering, based on cell temperature and retrieved water absolute amounts along with the relative difference between sample and calibration amounts, the
associated RCS corrections are minimised. The difference between sample and calibration retrieved $H_2O$ mole fractions (after QC/QA filtering) is -0.99 ppm ± 0.80. The difference in measured cell temperature between sample and calibration measurements is, prior to cell temperature sensor replacement, 0.04 °C ± 0.23 and after replacement, 0.08 °C ± 0.09.

We also decided to neglect flow rate RCS, more on theoretical grounds, as it induces a second order temperature effect. Changes in flow rate affect the measured cell temperature if the flowing gas is of a different temperature to the cell equilibrium temperature. Temperature distribution in the glass cell is also flow dependent (turbulent mixing). Prior to the decoupling of the cell pressure and flow, the flow was $0.53 \pm 0.03$ Lmin$^{-1}$. After the introduction of the duel MFCs, $0.50 \pm 0.005$ Lmin$^{-1}$, thus any potential flow RCS correction is minimal.

**5.7.1 Pressure residual cross sensitivity**

Pressure RCS (RCS$_p$) corrections need to be applied as cell pressure during sample and calibration measurements differ up to 100 hPa prior to cell pressure and flow decoupling (Fig. 4a). Experimental determination of RCS$_p$ is easily done. Experiments were repeated to assess long term stability of the RCS$_p$ in both modes of operation (static and flow) from 2009 to 2014. Table 1 lists the calculated RCS$_p$ for $CH_4$, CO and
$N_2O$. The derived values are consistent over a 5-year time span, over differing pressure ranges, sampling modes and pressure sensor calibrations. We expected RCS$_p$ to remain relatively constant as the pressure sensor has not been changed or relocated in the cell. In any such change, RCS$_p$ needs to be revaluated. Experimentally derived RCS$_p$ are in good agreement with H13, accept for CO which are of a magnitude less, this remains unexplained.

Figure 8a illustrates the calculated RCS$_p$ corrections applied to sample air dry mole fractions when using RCS$_p$ values of 0.034 ppb hPa$^{-1}$, 0.0009 ppb hPa$^{-1}$ and 0.005 ppb hPa$^{-1}$ for $CH_4$, CO and $N_2O$ respectively. After the decoupling of cell pressure and flow in April 2013 the sample and calibration cell pressures are effectively the same thus RCS$_p$ corrections are very small. The RCS$_p$ correction uncertainty is calculated by employing the ubiquitous propagation of error formulas (Ku,1966) using the uncertainty of the calculated RCS$_p$ and the
resolution of the pressure sensor. The associated uncertainties are displayed in Fig. 8b. The dominant component in the uncertainty is the RCS$_p$ uncertainty, not the pressure sensor uncertainty. We see calculated $CH_4$ RCS$_p$ correction uncertainty is of an order of magnitude less than the correction factor, but for $N_2O$ and CO comparable.





### 5.8 Measurement repeatability

As in G12 and H13 we quantify the precision of the FTIR in terms of measurement repeatability (GAW,2011). Successive repeatability tests over time are used to observe and assess changes in instrument precision. Such tests are an indication of measurement short term stability. Repeatability, over a given averaging time, is

calculated as the standard deviation of duplicate measurements of a gas sample of unaltering composition taken under constant conditions (i.e. cell pressure, cell temperature and cell flow rate).

Lauder FTIR repeatability experiments were performed by taking repeated 1-minute measurements of the TC under nominally standard unvarying operating conditions. Spectral analysis was conducted off-line to minimize

redundant time between measurements. The resultant species dry mole fraction time series were then analysed using the Allan variance technique (Allan, 1966) to characterise precision over differing temporal ranges. Figure 9 shows an example of the Allan deviations calculated from a repeatability experiment conducted in February 2015. For all three species, the Allan deviation (the square root of the Allan Variance) reduces with the square root of averaging time, consistent with being limited by Gaussian noise.

The base period for all sample and calibration measurements is 10 minutes, hence the 10-minute Allan deviation is taken as the operational instrument precision. The 10-minute Allan deviation per species from experiments conducted over 7 years of operation are listed in Table 2. $CH_4$ and CO 10-minute precision estimates of the Lauder FTIR are comparable to that reported in Griffith et al., (2011) but significantly less precise than that

reported in G12, especially $N_2O$. The design and operation of the Lauder FTIR is more comparable to the instrument used by Griffith et al. (2011) whilst the data used in G12 was acquired from a FTIR system (IUP, H13) more akin to the newer Spectronus design. The precision estimates are relatively stable over time for both measurement mode types. $CH_4$ and CO precision is well within the GAW recommended compatibility goals whereas the $N_2O$ precision is also close but does not meet the recommendation. For all three species, the

precision could be increased by extending the averaging time and/or replacement of the mid-IR detector with a more sensitive version. In the case of extending the averaging time, a balance must be found between a potential increase in precision and a small enough averaging time to capture short-term atmospheric variability.

### 5.9 Accuracy and Calibration

The spectroscopic retrievals and subsequent conversion to dry mole fractions are only as accurate as the underlying uncertainties associated with retrieval (i.e. forward model accuracy, spectroscopic linelist uncertainties) and inherent uncertainties of measured parameters (i.e., pressure and temperature sensor accuracy). Furthermore, the calculated dry mole fraction is not traceable to an absolute reference scale. MALT absolute accuracy is estimated to be approx. 2% (Griffith, 1996). This accuracy is not sufficient to meet the

intended purpose. Greater accuracy is achieved, along with mapping of the FTIR mole fractions to a known reference scale, by the ubiquitous method of measuring gases of known composition to derive an instrument response function (IRF). These gases are independently assigned, have high accuracy, and traceable to a defined international scale. When this method is applied, the overall accuracy of the FTIR is reliant on the calibration gas uncertainty, whereas precision is inherent in the FTIR itself.



From the measurements of the calibration gas an instrument response function (IRF) is constructed to map the retrieved dry mole fractions that of the assigned value. Such transfer functions are required for each species. The FTIR has been shown to have a linear response (H13) thus the IRF can be approximated by a first-degree (linear) polynomial, as in Eq. (1). $\chi_{ref\_meas}$ is the calibration gas dry mole fraction measured by the FTIR and $\chi_{ref}$ is the assigned calibration gas dry mole fraction. The IRF linear coefficients ($A_c$ and $B_c$) are derived using simple linear regression (using the ordinary least squares approach).

$$\chi_{ref\_meas} = A_c \chi_{ref} + B_c, \tag{2}$$

The air sample can then be calibrated as in Eq. (3), where $\chi_{cal}$ = calibrated sample amount, and $\chi_{rcs\_corr}$ is the air sample dry mole fraction after cross sensitivity correction. We see that when calculating the calibrated sample uncertainty, uncertainties associated with RCS corrections, WS assignment uncertainties, and to a lesser extent the derived IRF uncertainty need to be included.

$$\chi_{cal} = (\chi_{rcs\_corr} - B_c)/A_c, \tag{3}$$

The IRF linear coefficients are derived using a calibration suite with a minimum of two calibration tanks (of differing mole fractions), ideally three or more. The calibration suite composition should also span the range of expected atmospheric compositions. Unfortunately, the initial deployment of the FTIR at Lauder employed a single WS and continues to this day. This is sub-optimal, allowing only derivation of either the gradient or the intercept but not both simultaneously. To proceed, it is assumed that the IRF intercept ($B_c$) is zero, and the gradient ($A_c$) is to be calculated. This effectively reduces the IRF to a scale factor ($A_{sf}$). This approach will introduce a concentration dependent bias, this being the difference in $\chi_{cal}$ calculated using a scale factor (single point) calibration approach to that calculated using a full linear IRF parameterization.

The magnitude of the concentration dependent bias can be estimated by taking measurements of a multi tank calibration suite. First, the IRF is calculated from the multi tank suite in which both linear coefficients are calculated. We call this the Complete-IRF. Next, the IRF is derived using a single calibration tank (within the multi tank suite). This is called the scale factor. Combining Eq. (2) and Eq. (3) we can define concentration dependent bias in terms of the air sample dry mole fraction when calibrated using a single scale factor as in Eq. (4), where the concentration dependent bias is $\chi_c$-$\chi_f$. $\chi_c$ is the calibrated sample using the complete IRF and $\chi_{sf}$ is the calibrated sample using the scale factor.

$$\chi_c - \chi_{sf} = \chi_{sf}\left(\frac{A_{sf}}{A_c} - 1\right) - \frac{B_c}{A_c}, \tag{4}$$

Even given this limitation the use of a single scale factor for calibration still provides sufficient accuracy when calibration gas and air sample dry mole fractions are comparable (as shown in Sect. 5.9.3). The deficiencies in using single point calibrations are also encountered by Verhulst et al. (2017) in which concentration dependent bias is accounted for using a similar, but slightly different, methodology called extrapolation uncertainty.





The FTIR WS is dried ambient air collected at Baring Head during prevailing southerly winds, which is of comparable composition to Lauder baseline conditions (Saad et al., 2014). Due to the concentration dependent bias, only measurements taken in baseline conditions are currently used. Care should be taken in using the data

in conditions that are vastly different to the baseline conditions, such as night time inversion events.

### 5.9.1 Working standards

The working standards consumed by the FTIR are prepared and assigned at NIWA's greenhouse gas and isotopic analysis laboratory (NIWA-Gaslab) at Greta Point, Wellington. High pressure 30L aluminium cylinders (model 150A, Scott Marrin Inc., USA) are filled to approx. 2000 psig at Baring Head using a modified oil-free

compressor. During the filling process the air is also dried (<5 ppb) (Brailsford et al., 2012). WS assignment is then performed, using scale transfer reference gases on the current World Meteorological Organization (WMO) reference scales.

The composition and uncertainty of the WS used by the FTIR are listed in Table 3. One limiting factor of FTIR

accuracy is the uncertainty in the WS assignment. It is assumed that the tanks have a constant composition but in the majority of Lauder FTIR WSs there is significant drift in the CO concentration. It is vital that such drift be considered when scale factors are calculated.

### 5.9.2 Scale factor time series

As part of the standard operating conditions the WS was measured daily up until February 2014. After February 2014 weekly measurements were instigated. Figure 10 displays the calculated 7-day running mean scale factor for each species and associated uncertainties.  A 7-day running mean was used to minimise short term scale factor variability and provide a scale factor reproducibility estimate. The scale factors show that the inherent

accuracy MALT retrievals prior to calibration are better than 2% for $CH_4$ and 1% for $N_2O$, whereas up to 8% for CO.

Changes in the scale factor need to be accounted for. A step change is an indication an acute incident in the FTIR, FTIR acquisition procedure or a WS change. A gradual change indicates a change in FTIR performance or

WS composition drift. A change in the 7-day running mean scale factor standard deviation indicates a stability issue. A step change in the scale factor can be seen on WS change. This indicates a relative offset between the WSs, for example, the CO scale factor step change at the end of 2009 (Fig. 10c) indicates a possible mis-assignment of the WS and needs to be rectified. Any step change should be correlated with an instrument event (vertical dashed grey lines). For example, in mid-2011 there was an approx. 3% increase in the $N_2O$ scale factor

for a short period. This is associated with FTIR spectra acquisition using an input aperture of 3.0mm instead of 1.5mm. The two significant step changes in $CH_4$ and $N_2O$ in the 2010-2012 period are related the replacement of the temperature sensor and replacement of the FTIR internal metrology laser. There is an increase in the $CH_4$ scale factor standard deviation after 2014. This has been attributed to an error in the background spectrum $H_2O$





stripping procedure. This affects both sample and calibration measurements equally hence the calibrated sample measurements remain unaffected. Longer term gradual scale factor changes are harder to diagnose. The gradual decline in the $CH_4$ and $N_2O$ scale factors from 2007 to 2010 is unclear. Hypothesis include MIR globar intensity deterioration, cell wall effects and pressure/temperature sensor drift. The decline spans multiple WSs and instrument changes.

Drift in WS CO composition (Novelli et al., 1991; Andrews et al., 2014) is also a cause for concern and manifests itself in scale factor drift. If left uncorrected incorrect calibration of sample measurements occurs. Drift can be identified whilst the WS is in current use by a gradual increase in the scale factor, but only confirmed and quantified once the tank is returned to the calibration centre and remeasured. Thus, final sample calibration can only be achieved after WS re-measurement, hence in the interim all results are regarded as provisional. CO drift calculated after tank recalibration is listed in Table 3. CO drift is linearly parameterised and accounted for in the scale factor calculation. Figure 10c contrasts the scale factor calculated without drift correction (grey data points) and after drift correction (black data points). If drift correction is not taken into account, there will be an artificial downward trend in the calibrated sample CO measurements.

The scale factor uncertainty is calculated by combining the standard deviation of the 7-day running mean and the WS assigned uncertainty in quadrature. These can be viewed as the random and systematic components respectively. The total combined scale factor uncertainty are the black data points in Fig. 10. b,d,f and the uncertainty associated with WS assignment are the red data points. For $CH_4$ and CO, the WS assignment uncertainty is a significant component of the total scale factor uncertainty. With the instrument upgrade in April 2013 and changes in standard operating conditions in February 2014 there is a substantial reduction in the random uncertainty component resulting in total uncertainty being dominated by systematic uncertainty. Whilst there is a reduction in the overall $N_2O$ scale factor uncertainty due to the instrument and calibration procedure changes, the uncertainty related $N_2O$ measurement precision is still comparable to the WS assignment uncertainty. The spike in the $CH_4$ scale factor uncertainty starting in late 2013 coincidences with a reduction in the latency time within the calibration procedure. The abrupt uncertainty reduction in early 2014 is when the weekly flow mode calibration procedure started.

### 5.9.3 Multi-tank calibration suite measurements

A four-tank travelling set of scale transfer reference gases (collectively known as the Aniwaniwa suite) was purchased in 2014. The suite composition matrix was designed with the FTIR in mind. Details on the Aniwaniwa suite can be found in Appendix C. Primarily, the Aniwaniwa suite is to provide independent travelling standards for the New Zealand carbon monitoring network to assess site to site bias. It is also used as an independent assessment of the FTIR Complete-IRF. This is done by comparing uncalibrated (but cross sensitivity and water corrected) FTIR measurements of the suite against the suite assignments. From this the Complete-IRF can be calculated. Conversely, by calibrating the suite measurements using the single scale factor (the same method used to calibrate sample data) and then comparing to the suite assigned values WS bias can be





diagnosed. Aniwaniwa measurements also allows investigation into the concentration dependent bias arising from using a single calibration tank.

The Aniwaniwa suite is intended to be measured at Lauder at regular intervals, so far only twice, once in November 2014 (N14) and again in November 2015 (N15). The FTIR measurements are made using the same procedure as to that of regular WS and TC measurements. The Aniwaniwa suite, WS and TC tanks are measured each in turn for 10 minutes, in flow mode for 1 hour. This is then repeated. Overall, 60 L of gas per tank is consumed.

In addition, in April 2010 a GAW performance audit of Lauder was conducted by World Calibration Centre at the Swiss Federal Laboratories for Materials Science and Technology (WCC-EMPA, Zellweger et al., 2010). As part of the audit activity the 6 tank WCC-EMPA travelling standard suite was measured by the FTIR. Measurements of this additional multi-tank suite are also used to assess the FTIR IRF stability in an earlier period of the FTIR operation prior to the Aniwaniwa suite purchase. The measurements were made with a similar methodology of that used to measure the Aniwaniwa suite.

Table 4 lists the Complete-IRF coefficients calculated from the three-suite measurement sets and Fig. 11 shows the residual fits of the Complete-IRF per species. Since only three multi tank sets have been measured in the past eight years conclusive results cannot be drawn, but given the time span, they still offer an indication of the FTIR IRF stability and linearity. Across all species, the coefficients calculated from the N14 and N15 measurements are in close agreement indicating good stability over a year of operation. The coefficients derived from the WCC measurements in 2010 (W10) are in less agreement with the N14 and N15 values. One reason for this difference is that the W10 measurements were made prior to the April 2013 upgrade. Prior to the upgrade tank gas was not dried by the FTIR system, hence water vapour varied between tank measurements of up to 20 ppm.

As in the calculation of the WS scale factors, we expect to see changes in the Complete-IRF with changes in the instrumentation. In any implementation of a Complete-IRF in routine sample calibration will still require regular measurements of a multi-tank calibration suite either by employing external scale transfer reference gas suites or a suite of multiple WSs (of differing composition). The latter option is preferable. Also of note, the associated uncertainties in all sets (N14, N15 and W10) are similar, another indication that instrument precision has not degraded over time.

In the next application we calibrate the suite measurements in the same manner as sample data, by applying a recent calculated scale factor. Comparing the assigned W10, N14 and N15 suite tank values to the difference between the calibrated measurement and assigned values (Fig. 12) gives an insight to the magnitude of the concentration dependent bias arising from the use of a single WS calibration procedure. The dash-dot-dot lines in Fig. 12 are the calculated concentration dependent biases for each suite measurement set. For all three species the calculated concentration dependent bias gradient and measurement assigned difference gradients are similar, indicating that concentration dependent bias is the main cause of the increasing discrepancy between calibrated measurements and assigned amounts with increasing concentration. The offset between the concentration





dependent bias and the measurement-assigned difference is a result of bias between the FTIR WS and the tank suite assignments. The concentration dependent bias is minimal for all species over the baseline range (grey shaded area in Fig. 12.) and comparable to GAW compatibility goals at higher mole fractions. The concentration dependent bias is also smaller than the seasonal cycles and annual trends seen at Lauder (see Sect. 8 for trend

analysis) so we have confidence that the concentration dependent bias introduced using the scale factor calibration method will not affect baseline data analysis.

Of more concern is the large positive bias of the calibrated $N_2O$ FTIR measurements. The probable cause is that for $N_2O$ NIWA-Gaslab use synthetic composition scale transfer references gases. A 0.65ppb bias was observed

in WCC-$N_2O$ travelling standard measurements at NIWA-Gaslab during an audit of the Baring Head GAW station in 2009 (Scheel, 2012). This is a similar bias to what is seen in N14 and N15 measurements. Even if this offset is taken in account a bias will remain (in the region of 0.35 - 0.7 ppb). This offset will not introduce a bias between the FTIR and flask sample measurements as both datasets are calibrated using WSs made and assigned at NIWA-Gaslab. The bias will need to be addressed before the Lauder $N_2O$ FTIR (and flask) measurements can

be used in conjunction with other institute's datasets apart from trend analysis comparison.

 The consistency of the $CH_4$ measurements across all three sets indicate a stable IRF and consistent WS assignment. For CO concentration dependent bias is evident, but only significant outside baseline conditions. N14 and N15 concentration dependent bias have comparable gradients, but offset, indicating a small relative

mis-assignment between consecutive FTIR WSs. The $N_2O$ concentration dependent bias is relatively small compared to the overall bias.

### 5.10 Measurement Reproducibility

The series of repeatability experiments over 2008-2015 provide snapshots of instrument short term stability.

Assessing instrument reproducibility over longer time scales requires a different approach.

The approach we take is to take regular measurements of a target cylinder. Theoretically, repeated measurements taken in the same conditions should give the same results. Measurement spread allows us to quantify instrument reproducibility and assist diagnosis of instrument changes or faults.

Target cylinders are prepared and assigned at NIWA-Gaslab in the same manner as WSs. We found composition assignment advantageous (but not critical) in that we can also quantify the measurement bias hence accuracy can also be regularly ascertained. Without knowing the composition then only the reproducibility can be assessed.

Routine TC measurements started in August 2013, with sporadic measurements prior to that. A single TC is measured in the same manner as that of the WS. A total of 1322 days of TC measurements were taken (2010 - 2017). Daily static mode TC measurements were taken up to Feb 2014. When calibrations switched to weekly flow mode measurements so did the TC measurements. Figure 13 shows the difference between the TC dry mole fractions measured by the FTIR and that of the TC assigned values. Vertical dashed lines indicate a change in





tanks (WS or TC) or major instrument change. Within these stable intervals, the standard deviation of the TC measurements is an indication of instrument reproducibility whilst inter-interval difference indicates a systematic bias attributed to the event causing an interval change. TC measurement bias and standard deviation in each interval, and for the total dataset, are listed in Table 5. For all three species reproducibility estimates are of greater value then precision estimates indicating small changes in standard operating conditions affect measurements. $CH_4$ and CO reproducibility is within the GAW recommended compatibility goals, whilst $N_2O$ is nearly double. Across all intervals, the measured to assigned differences are remarkably Gaussian in distribution given the intra-interval systematic differences. The exception is CO, in which interval C, D and E biases dominate (Fig.13d) indicating possible issues in WS assignment. In all intervals, for all species, reproducibility estimates are within the GAW compatibility recommendations and small enough to allow statistically significant annual trend and seasonal cycle analysis.

The interval TC differences can be used to assess the effects of instrument changes and identify possible issues with both TC and WS assignments. For example, after a WS change intervals E and F have a CO bias of approx. 1.7 ppb, which is greater than the combined reproducibility of both intervals. Given that TC measurements in intervals C, D & E are all high, this indicates the WS assignment used in these periods need to be scrutinised. Conversely, on the change of the TC over intervals H to I the bias is approx. 0.5ppb indicating possible TC assignment issues. Considering that both the TC and WS are prepared in the same laboratory, using the same method, there should be no systematic differences between tank assignments. Furthermore, WS and TC compositions are similar as both tanks are handled, measured and analysed the same way on the FTIR.

H13 assessed the reproducibility of the IUP FTIR over a period of 6 months and reported values of 0.28 ppb, 0.45 ppb and 0.1 ppb for $CH_4$, CO and $N_2O$ respectively. On an interval by interval basis the Lauder FTIR reproducibility is comparable to H13 for CO, but near double that for $CH_4$ and $N_2O$. The greater variance cannot be explained by WS or TC assignment uncertainty as the analysis is within each interval, and the interval span is similar in length to the 6-month measurement period in H13. This indicates there still is room for improvement in the measurements at Lauder, such as mid-IR detector (better SNR) and White Cell upgrades (better thermal stability and cell gas mixing).

### 5.11 Data quality assurance and quality control (QC/QA)

Very little has been explicitly published on FTIR QC/QA schemes. At Lauder, two filtering methods are used to exclude questionable data from the dataset. The first method is an objective diagnostic filtering scheme in which data are rejected based upon spectral processing diagnostics and cell state parameters. No filtering is performed on species dry mole fractions, only $H_2O$ is filtered upon. Table 6 presents the list of the diagnostics that are filtered upon, and threshold limits. The threshold limits are empirically set, based upon standard operating conditions at Lauder. The threshold limits have been set to exclude outliers (approx. 3σ). Acquisition software upgrades in April 2013 enabled the recording of cell pressure, cell flow and cell temperature standard deviations within a single measurement averaging period. These were added to the list of diagnostics and allows filtering upon FTIR stability within a single measurement. The difference between successive measurement diagnostics



(rate of change) such as H$_2$O concentration, cell pressure and cell temperature are used to filter out any short-term instrument changes (e.g. on chemical desiccant change or instrument restart).

Objective filtering cannot capture all instances and a second method based upon user defined date/time periods
to omit data is also used. Such manual filtering is subjective and reliant on the operator to identify and record these periods. Examples of such, are data taken during instrument testing, instrument component failure/leaks and external events that could influence measurements such as farm machinery operating close to the sampling inlet (i.e. enhanced CO). An event log is kept and updated at regular intervals. Changes to FTIR hardware, operating conditions or analysis are recorded. Step changes in instrument performance or analysis should align
with these recorded events and can be used to set up manual filtering to omit data.

Overall, between 2007 to 2017 approx. 423,000 10-minute atmospheric sample measurements had been taken, 88% pass the objective filtering threshold limits, then reducing to 80% after manual filtering (there were 93 specific manual filtering intervals). The main causes of manual filtering omission are instrument repairs, testing
and instrument upgrades. There was a prolonged period (2009.0-2011.2) in which there was increased data rejection (Fig. 14). The reason was incomplete filling of the cell during the first sample measurement after the daily calibration cycle. A blockage in the chemical desiccant cartridge reduced cell fill rate.

**5.12 Calibrated CH$_4$, N$_2$O and CO air sample timeseries**

Figure 14 displays the entire filtered time series of calibrated CH$_4$, CO and N$_2$O dry mole fractions at Lauder.
Measurements taken in baseline conditions are highlighted in red. From this we see the large enhancements are outside baseline conditions, primarily at night where build up is seen in the nocturnal boundary layer. The isolated large spikes of CO are due to local farmland prescribed burns.

The calibrated sample measurement uncertainty is calculated by combining the sample measurement precision
(Table 2), scale factor uncertainties (Fig. 10) and RCS$_p$ correction uncertainties (Fig. 8a) using standard error propagation methodology (Ku, 1966) in a manner similar to that used by Verhulst et al. (2017). Furthermore, the uncertainties can be grouped into systematic (RCS$_p$ corrections, WS uncertainties) and random (scale factor 7-day running mean standard deviation and sample measurement precision) components.

Figure 15 displays the total, systematic and random uncertainties of the calibrated timeseries for each species. The average uncertainty is approx. 1.4 ppb, 0.6 ppb, and 0.3 ppb for CH$_4$, CO and N$_2$O respectively, with uncertainty proportional with measurement concentration (due to error propagation). The short duration large spikes in uncertainty are related to instances of high sample measurement concentrations in which uncertainties propagate. For two instances in the CH$_4$ record (at the start of 2007 and 2014) the large uncertainty is due to a
larger than usual scale factor uncertainty. Since the upgrade in April 2013 RCS$_p$ corrections for all species have been negligible, hence a reduction in associated uncertainty. Overall, the CH$_4$ total uncertainty has remained constant across the time series with total uncertainty dominated by the WS uncertainty. The reduction in the CH$_4$ random uncertainty at the end of 2012 is due to revised precision estimates, and further reduction in random uncertainty post 2014 is due to the combination of the April 2013 upgrades and the change to flow mode



calibration measurements. For CO, random and systematic uncertainty components are similar in magnitude. From 2007 – 2010, there was a small downtrend in the CO random uncertainty component (approx. 0.1ppb over 7 years), which cannot be fully explained by application of revised precision estimates (Table 2), Such revision of estimates can explain the stepwise reduction in early 2015. There is a pronounced seasonal cycle in the CO

systematic uncertainty, more so than for $CH_4$ and $N_2O$, as there is a approx. 20% seasonal cycle in atmospheric CO observed at Lauder. For $N_2O$, up until the April 2013 upgrade, the random component of the total uncertainty was greater than the systematic component indicating instrument precision was a limiting factor. After the upgrade, there was a reduction is the scale factor uncertainty with systematic and random components now being comparable in magnitude.

**6. Flask sample measurements and analysis**

Routine (weekly) in situ flask air sample collection at Lauder started in May 2009 as a robust proven cost-effective approach to provide independent measurements of $CH_4$, CO, $N_2O$, $CO_2$ and $\delta^{13}C$-$CO_2$ for comparison against FTIR measurements. Flask samples will also assist in identifying any issues or artefacts arising from the air sampling system. TC measurements cannot do this. We have also used FTIR measurements to help identify

issues in flask measurement and analysis, hence such comparisons provide a two-way check.  One drawback of flask sampling is that measurements are not continuous, offering only a sparse temporal dataset. We decided to only collect air samples in baseline conditions as this is when atmospheric composition is varying least. This assists in reducing concentration differences arising from differences in instrument sampling time and duration.

NIWA has a long term in situ flask sampling programme at Baring Head and Arrival Heights, Antarctica (77.82 S, 166.65 E, 220m AMSL) (Lowe et al., 1994). The samples collected at Lauder follow the same collection methods and laboratory analysis. In brief, air is drawn from the air sampling manifold at a rate of 2.0 $Lmin^{-1}$ through 5 metres of nylon tubing (model N12-04 series 1200 Ledalon, New Zealand) with a diaphragm pump (model N86KTE, KNF Neuberger, Germany). An inline magnesium perchlorate cartridge is used to dry the

sample air (effective dew point of approx. -60.0 °C) before reaching two evacuated glass 2.2L sampling flasks (Glasscraft Scientific Glass-blowing Limited, New Zealand). These two flasks are attached in parallel. The flasks are flushed five times with sample air to a pressure of 20 psig, after which the flask is filled to 20 psig. Final filling time is approx. 5-8 minutes. The magnesium perchlorate cartridge is replenished after 12 sample collections. Over the 2009-2017 period there has been no alterations in either the collection procedure or flask

sampling system.

Analysis of the flask air is performed at NIWA-Gaslab. Gas chromatography (GC) flame ionization detector, GC cold vapor atomic fluorescence spectrometry, and GC electron capture detector laboratory techniques are used to determine the dry mole fraction content of flask samples for $CH_4$, CO and $N_2O$ respectively. The WMO

reference scale used to assign the FTIR WS and TC are used in the analysis of the flask samples. The paired flask samples are a quality assurance measure. Samples with intra-flask differences greater than the combined uncertainty in each sample pair are rejected or if flask difference exceed 2.0 ppb, 1.0 ppb and 0.5 ppb for $CH_4$, CO and $N_2O$ respectively. These limits are based on the GC technique measurement uncertainty. Rejected





samples indicate either a failure in collection or GC analysis. The mean value of the flask pair, along with the combined individual flask uncertainties, is used in comparisons with FTIR measurements.

**7. FTIR flask sample comparison**

Comparison between FTIR measurements and flask samples are conducted on a regular basis. This achieves two objectives, assessing flask data quality and to check if there is any change in the bias between the two measurements. Change in bias indicates either a fault (or drift) in one (or both) of the measurements which needs to be investigated and accounted for.

Flask sample filling time is 5-8 minutes, offering only a snapshot of atmospheric composition, whereas the FTIR 10-minute measurements are continual, with an e-folding time of approx. 7 mins. Since the FTIR individual measurements are not independent of each other, comparing a single FTIR measurement with a flask sample measurement is not straight forward. To minimise such temporal effects, flask measurements are taken in baseline conditions and compared to 1-hour FTIR averages, which consists of 6 measurements. This also provides an estimate of baseline variability. The integrating effect of the different measurement sample volumes, as applied by Winderlich et al. (2010), has not been employed in this analysis due to minimal baseline variability. This approach would need to be used when comparing the measurements taken in conditions of high variability (i.e. during nocturnal boundary layer inversion events).

Figure 16 displays the FTIR flask comparison results for $CH_4$, CO and $N_2O$. Table 7 lists measurement dataset biases along with the results from simple linear regression (using the ordinary least squares approach) of the FTIR against flask. The total uncertainty in the FTIR flask difference is the uncertainty in the flask measurement added in quadrature with the FTIR measurement uncertainty. The recommended GAW compatibility goals are also displayed to assist in interpretation (and add perspective) of the differences against an international standard. As illustrated in all time-series (Fig. 16 a, d, g) there is a gap in comparison sample points between mid-2009 to mid-2010. This is due to two factors, mainly flask samples being taken outside the defined baseline criteria and, to a lesser extent, flask samples not passing quality assurance checks. It was only in mid-2010 that we decided focus on taking all samples during baseline conditions thus the effective comparison period is better defined as mid-2010 through to mid-2015.

The $CH_4$ FTIR flask comparison results show good agreement between the two measurement datasets with a bias of -1.02 ppb ± 2.61. Differences are not concentration dependent, show a tight linearity and are not seasonally dependent. The larger differences seen in the period 2014.5 to 2015.5 are not accounted for, requiring further investigation. The $CH_4$ bias and standard deviation is comparable to other FTIR comparison activities (Griffith et al., 2011; Vardag et al., 2014) and comparable to other continuous $CH_4$ measurement techniques in comparison to co-located flask measurements (Winderlich et al., 2010; Popa et al., 2010).

The CO FTIR flask comparison results show a tight linearity and are not seasonally dependent. The bias of -0.43 ± 1.60 ppb indicates an overall good agreement between measurements and are within the GAW recommended





compatibility target of 2 ppb. The CO bias and standard deviation is akin to results from other continuous CO instruments compared to co-located flask measurements (Thompson et al., 2010; Popa et al., 2010).

For $N_2O$, a bias of -0.01 ± 0.77 ppb is within the GAW recommended compatibility goal of 0.1ppb but this is more serendipitous when the FTIR flask time series and correlation scatter plots are viewed (Fig. 16g, h). Care must be taken in interpretation as systematic differences dominate in different time periods, but as an ensemble, produce statistical results that could convey a large, but Gaussian spread (Fig. 16i). For instance, there is an increased bias over the time interval 2014.65-2016.08. So far, the causes are unknown. There is no explicit correlation between the bias with any FTIR instrument or flask sample events, and only affects $N_2O$ (not CO or $CH_4$). We suspect the issue is with the FTIR measurement as the elevated level of $N_2O$ is greater than what simple trend analysis would indicate, as seen in the baseline time series (see Fig. 17c). There is also a sudden (step) decrease of $N_2O$ at the start of 2016 that is not seen in the $N_2O$ flask samples. $N_2O$ FTIR comparison measurements carried out by Griffith et al., 2011 show much better results. A bias of -0.12 ppb was also reported but with a standard deviation of 0.22 ppb. $N_2O$ FTIR comparisons conducted by Vardag et al. (2014), also report a much smaller standard deviation (0.22 ppb) than our results. A comprehensive investigation of five continuous $N_2O$ analysers (including the FTIR) by Lebegue et al. (2016), showed FTIR performance comparable to the other instruments. These findings point to a specific but as yet unidentified issue with the Lauder FTIR $N_2O$ measurements. It also highlights the need for independent dataset validation, as internal FTIR QC/QA did not identify any issues over the 2014.65-2016.08 period.

Flask sampling will continue at Lauder. The next step is to collect flask samples outside baseline conditions allowing an independent check against FTIR measurements taken in conditions with higher mole fractions and variability, such as during nocturnal boundary layer conditions. Such data will also provide an independent dataset to assist in assessment of concentration dependent bias, arising from the use of a single WS for calibration.

## 8. FTIR baseline measurement time series analysis

Here we perform and present baseline timeseries trend analysis. We focus on baseline measurements as they are representative of the regional atmosphere, minimally affected by local emissions and conditions, simpler to interpret and give a better indication of any instrument fault or change. We want to see if the FTIR measurements are sufficiently accurate, precise, stable and reliable enough to capture annual and seasonal changes. These are the most trying conditions to measure over the longer term. Analysis and commentary on diurnal cycles and night time measurements are outside the scope of this work.

The trend analysis technique used by Gardiner et al. (2008) was applied to the FTIR baseline datasets. Residual resampling (boot strapping) using 5000 iterations was performed. A linear fit (a broad simplification) and single Fourier pair was used in trend analysis for $CH_4$, CO and $N_2O$. A single Fourier pair was sufficient to capture the seasonal cycle. There was no substantial reduction in the goodness of fit with additional Fourier components. Additional Fourier terms also complicate physical attribution interpretation. This simplistic linear and single Fourier pair approach is sufficient for the objectives we are trying to achieve in this analysis.





Figure 17 displays the baseline time series of all three species. Qualitatively, $CH_4$ and $N_2O$ measurements exhibit an increase over time, whereas CO shows a minimal decrease. As expected, the baseline flask samples also display similar patterns. The detrended seasonal cycles are displayed in Fig. 18. Table 8 lists the trend analysis results, peak-to-peak seasonal cycle amplitudes and associated uncertainties. The bootstrap bias correction index (Efron and Tibshirani, 1993; Gardiner et al., 2008) of 0.47, 0.49 and 0.52 (for CH4, CO and N2O respectively) indicates the analysis method does not introduce significant bias.

The annual linear trend in methane of 6.29 ppb year$^{-1}$ since 2007 is consistent with other southern hemisphere mid latitude in situ measurement studies (Nisbet et al., 2016; Dalsøren et al., 2016.). The observed peak-to-peak seasonal cycle amplitude of approx. 29 ppb (peaking in winter time) is dominated by OH oxidation and is consistent with current understanding (Dlugokencky et al., 1997). There is greater variability and elevated amounts in the spring-summer time measurements; evidence of possible local horticulture and agriculture emissions and/or seasonal transport of enriched $CH_4$ air from other regions The explanation of the causes is outside the scope of this work.

CO has a linear annual trend since 2007 of -0.52 ppb year$^{-1}$. The measurements agree with other southern hemisphere in situ measurements (Zeng et al., 2015) and the observed trend is like that derived from remote sensing measurements of CO at Lauder (Zeng et al., 2012). The seasonal cycle is also in agreement with measurements made at Cape Grim, Australia (40.68 S, 144.68 E, 91m AMSL) (Fisher et al., 2015). This is not surprising as both as sites are in the southern mid latitudes and described as remote clean air stations.

Baseline $N_2O$ data over the period 2014.65-2016.08 are not used in seasonal cycle and trend analysis due to possible FTIR instrument issues (see Sect. 7 for details). To check, the annual trend calculated with inclusion of the flagged erroneous data was estimated at 1.06 ppb year$^{-1}$ (± 0.01) compared to 0.99ppb year$^{-1}$, demonstrating that inclusion alters the trend estimate by approx. 6%. The linear annual trend of approx. 1ppb year$^{-1}$ (0.3% year$^{-1}$) is similar to annual growth rates deduced from measurements over 2008-2012 made at Baring Head (0.17 - 0.32% year$^{-1}$) and Cape Grim (0.3 - 0.36% year$^{-1}$) (Ye et al., 2016). The reduction in $N_2O$ concentration in 2009 is thought to be real, not an instrument artefact, as a decline is also seen in these other site measurements (Ye et al., 2016). The bootstrap analysis technique indicates there is a small but statistically detectable peak-to-peak seasonal cycle amplitude of 0.52 ppb, approximately double that of the instrument reproducibility (0.27 ppb).

**9. Summary**

Operation of the FTIR over 10 years has shown the instrument repeatability and reproducibility to be stable over the long term, even across significant instrument and analysis upgrades. The Bruker IRcube has shown to be reliable with a stable ILS producing spectra with high SNR. Neither ILS stability nor SNR are limiting factors in measurement uncertainty. The FTIR is of high reliability. Component failure is rare. Consumables (i.e. dry



nitrogen, desiccant and the metrology HeNe laser) are easily replaced. Operator intervention (to perform tests and upgrades) along with desiccant replacement are the main causes of data collection interruption.

Changes in the cell temperature sensor placement, and type, have increased temperature monitoring precision and are now more responsive and representative of cell gas temperature. The instrument upgrades in April 2013 decoupled and increased control over cell pressure and cell flow rate resulting in a significant reduction in pressure residual cross-sensitivity corrections. Pressure residual cross-sensitivity experiments spanning approx. 5 years are in good agreement, again indicating FTIR measurement stability across multiple upgrades and changes.

Introducing a new CO and $N_2O$ MALT retrieval strategy has significantly reduced CO and $N_2O$ cross-sensitivity to $^{12}CO_2$. There is also an added benefit in that the $N_2O$ pressure cross sensitivity can now be represented as a linear function. The addition of the background spectrum water stripping procedure produces a transmission spectrum that can be more realistically modelled reducing retrieval uncertainty (hence an increase in reproducibility).

Instrument precision experiments spanning multiple years are within GAW recommended compatibility goals for $CH_4$ and CO and comparable for $N_2O$. Instigation of target cylinder measurements allows diagnosis of medium term (months to years) reproducibly and, if the tank has an assigned value, it can be used to investigate calibration accuracy.

By using a single WS to calibrate samples, concentration dependent bias is introduced but the effect is minimized when the WS composition is akin to that of sample air. A multi tank reference suite with a custom composition matrix tailored for the FTIR was constructed. Annual measurements of the Aniwaniwa suite, along with the WCC-EMPA audit suite show FTIR instrument response function is stable and the concentration dependent bias (arising from single WS calibrations) is minimal. Measurements of the Aniwaniwa suite also allow inference of WS accuracy. This cannot be deduced solely from the FTIR flask comparisons as FTIR WS assignments are measured using the same laboratory techniques and scale transfer standards as that used in flask analysis.

An uncertainty budget for calibrated sample measurements was constructed and decomposed into random and systematic constituents. The April 2013 instrument upgrades reduced $CH_4$ random uncertainty, so that systematic uncertainty now dominates $CH_4$ total error. The upgrades also reduced CO and $N_2O$ random uncertainty but are still comparable to that of systematic uncertainty.

Comparison of FTIR and co-located flask measurements show good agreement for $CH_4$ and CO. Whilst the bias of $N_2O$ FTIR flask comparisons is within GAW recommended compatibility goals, this is serendipitous and dominated by systematic differences. Simplistic baseline time series trend analysis was conducted with calculation of linear annual trends and seasonal cycles. The deduced trends and seasonal cycles align with estimates from other southern hemisphere in situ measurements.




Apart from one inconclusive study, there is a lack of FTIR CO comparison activities with other co-located measurements. Whilst the results of this study indicate FTIR CO measurements meet GAW reproducibility and compatibility recommendations, we recommend additional comparisons especially against other continuous in situ instruments.

Improvements can be made in many areas of operation. Upgrading the Mid-IR detector and White cell, to those used in the commercially available Spectronus FTIR systems, would increase spectra SNR and cell thermal stability respectively. These two changes will ultimately lead to an overall improvement in measurement repeatability (and reproducibility). A more sophisticated desiccant replacement system would reduce measurement down time, or pre-flushing the newly refilled trap with dry air or nitrogen from a tank before installing it inline. Using multiple WS (of differing composition) would eliminate concentration dependent bias, hence providing increased accuracy of measurements outside baseline conditions. Multiple WSs would also allow more timely analysis of drift in tank composition (especially CO). Flask samples should also be taken outside baseline conditions over a greater composition comparison range. This would help diagnose the extent of the concentration dependent bias, when using a single WS for calibration.

As the Aniwaniwa and WCC-EMPA measurements show, the large positive bias of calibrated $N_2O$ measurements needs to be reconciled before the data can be used in conjunction with other institutes' datasets. Despite these misgivings, the current FTIR system employing single WS calibrations is sufficient to capture $CH_4$, CO and $N_2O$ seasonal and annual trends in southern hemisphere atmospheric composition within GAW reproducibly guidelines. Calibrated and quality controlled $CH_4$ data have already been submitted to the GAW World Data Centre for Greenhouse Gases database (GAW ,2009) and submission of CO is planned.

**Data availability**

Calibrated baseline $CH_4$ FTIR and $CH_4$ flask sample measurements can be found at: https://ds.data.jma.go.jp/gmd/wdcgg/cgi-bin/wdcgg/accessdata.cgi?index=LAU545S00-NIWA&select=inventory. Other data in this publication can be obtained from the corresponding author on request.

**Author contribution**

DS is responsible for FTIR operation, data management, data QC/QA and data calibration. DG developed the FTIR analyser, and MALT retrieval code. DS, DG, GB and MK initially installed the FTIR at Lauder. DS developed the air sampling system. DS, VS and DG designed experiments and associated analysis techniques. WS and TC construction and assignments were conducted by GB, RM and SN. Flask measurements were taken by MK and DS with subsequent analysis conducted by GB, RM and SN. DS prepared the manuscript with contributions from DG, SN and RM.





**Competing interests**

The Lauder FTIR analyser was built at UoW in 2006 before the involvement of Ecotech P/L in the further development of the analyser as Spectronus (from 2011). DG and UoW have since operated as consultants to Ecotech.

**Acknowledgements**

The Lauder in situ greenhouse gas monitoring project is supported by NIWA core funding through funding from New Zealand's Ministry for Business, Innovation and Employment. We would like to thank to Nicholas Deutscher, Stephen Parkes, Martin Riggenbach, Sam Hammer and Graeme Kettlewell for invaluable technical advice and support. We would also like to thank Duane Kitzis at NOAA ESRL GMD for the Aniwaniwa suite
construction and Christoph Zellweger for the WCC-EMPA 2010 Lauder station audit. Finally, a big thanks to Sara Mikaloff-Fletcher for helpful comments and advice.

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





## Tables

Table 1. CH₄, N₂O and CO RCSₚ including values from H13 (1σ uncertainty in brackets). The date of experiments is
given in the first column. In the second column are the pressure ranges and steps (bracketed) the experiments were
10  conducted at. The experiment measurement mode is listed in the last column.

| Date | Pressure range Low-High (step) [hPa] | CH₄ RCSₚ [ppb hPa⁻¹] | CO RCSₚ [ppb hPa⁻¹] | N₂O RCSₚ [ppb hPa⁻¹] | Mode |
|---|---|---|---|---|---|
| March 2009 | 650-950 (50) | 0.023 (0.002) | 0.004 (0.001) | - | static |
| Nov 2011 | 730-1050 (20) | 0.030 (0.001) | 0.0013 (0.0008) | 0.003(0.002) | flow |
| Aug 2012 | 750-1100 (50) | 0.033 (0.001) | 0.0006 (0.0001) | 0.005(0.001) | static |
| Jan 2013 | 650-1050 (25) | 0.030 (0.002) | 0.0005 (0.0002) | 0.004(0.001) | static |
| Dec 2013 | 650-1050 (25) | 0.030 (0.002) | -0.0006 (0.0001) | 0.005 (0.0008) | flow |
| Jan 2014 | 800-1200 (50) | 0.034 (0.002) | 0.0009(0.0018) | 0.008 (0.0013) | flow |
| H13 | 800-1200 | 0.031 (0.003) | 0.006 (0.002) | 0.007 (0.001) | flow |

15  Table 2. CH₄, CO and N₂O 10-minute Allan deviation estimates measured at Lauder along with estimates from
Griffith 2011 and G12. The GAW recommend compatibility goals are also listed for comparative purposes. The
measurement mode and number of 1-minute data points (N) used in each Allan Variance analysis experiment are
listed.

| Date | CH₄ [ppb] | CO [ppb] | N₂O [ppb] | Mode | N |
|---|---|---|---|---|---|
| Apr 2008 | 0.64 | 0.38 | 0.23 | static | 440 |
| Apr 2010 | 0.66 | 0.35 | 0.24 | flow | 440 |
| June 2012 | 0.23 | 0.31 | 0.11 | flow | 280 |
| Nov 2012 | 0.28 | 0.30 | 0.10 | static | 1000 |
| Nov 2012 | 0.19 | 0.31 | 0.13 | flow | 170 |
| Jan 2014 | 0.25 | 0.28 | 0.13 | flow | 450 |
| Feb 2015 | 0.40 | 0.21 | 0.11 | flow | 170 |
| Griffith et al. (2011) | 0.2 | 0.2 | 0.06 | flow | |
| G12 | 0.06 | 0.08 | 0.03 | flow | |
| GAW compatibility | 2.0 | 2.0 | 0.1 | | |



goals

**Table 3. Working standards consumed by the FTIR. WS CH₄, CO and N₂O dry mole fraction assignment with 1σ uncertainty bracketed. Working standard tank date of attachment to the FTIR, tank identifier and calculated CO drift rates are given. The CH₄, CO and N₂O assignments are traceable to the WMOx2004A (Dlugokencky et al., 2015), WMOx2014A (Novelli, et al. 1991) & WMO2006A (Hall et al., 2007) reference scales respectively.**

| WS identifier | Date | CH₄ [ppb] | CO [ppb] | CO drift [ppb year⁻¹] | N₂O [ppb] |
|---|---|---|---|---|---|
| REF6026 | Jan 2007 | 1709.81 (1.03) | 48.15 (0.55) | 0 | 318.75 (0.17) |
| REF13416 | Apr 2008 | 1733.28 (1.05) | 56.85 (0.60) | 0 | 319.49 (0.21) |
| REF6955 | Nov 2008 | 1751.90 (0.66) | 63.01 (0.55) | 0 | 320.92 (0.10) |
| REF7193 | Apr 2010 | 1779.72 (0.96) | 68.90 (0.23) | 0.42 | 322.57 (0.13) |
| REF9580 | Dec 2012 | 1752.01 (1.23) | 53.67 (0.77) | 3.13 | 323.61 (0.09) |
| REF12510 | Nov 2013 | 1769.38 (1.08) | 58.05 (0.40) | 1.85 | 325.16 (0.16) |
| REF13009 | Sept 2014 | 1799.81 (1.24) | 69.46 (0.21) | 1.55 | 326.73 (0.15) |
| REF13486 | June 2016 | 1797.40 (1.60) | 55.50 (0.20) | 0.76 | 328.50 (0.20) |

**Table 4. The Complete-IRF gradient and intercept coefficients for each species calculated from three suite measurements sets (1σ uncertainty in brackets). W10 is the WCC-EMPA travelling standard suite measured in 2010. N14 and N15 are the measurements of the Aniwaniwa suite in 2014 and 2015 respectively. The coefficient of determination (r²) of each fit is supplied.**

| Suite ID | CH₄ gradient [ppb ppb⁻¹] | intercept [ppb] | r² | CO gradient [ppb ppb⁻¹] | intercept [ppb] | r² | N₂O gradient [ppb ppb⁻¹] | intercept [ppb] | r² |
|---|---|---|---|---|---|---|---|---|---|
| W10 | 1.015 (0.002) | -12.17 (4.21) | 0.999 | 1.046 (0.008) | -1.46(1.01) | 0.999 | 1.002 (0.013) | -2.64 (3.92) | 0.999 |
| N14 | 1.021 (0.002) | -5.98 (4.04) | 0.999 | 1.057 (0.009) | 1.58(0.87) | 0.999 | 1.011 (0.009) | -0.82 (3.00) | 0.999 |
| N15 | 1.021 (0.002) | -7.09 (3.33) | 0.999 | 1.061 (0.007) | 2.31(0.78) | 0.999 | 1.011 (0.011) | -1.19 (3.80) | 0.999 |

**Table 5. For each interval (and total dataset) in fig. 13, the measured to assigned TC bias is listed. Interval reproducibility (1σ standard deviation) is bracketed. N = total number of TC measurements per interval.**

| Interval | CH₄ bias [ppb] | CO bias [ppb] | N₂O bias [ppb] | N | Interval length (days) |
|---|---|---|---|---|---|
| a | -0.36 (0.78) | 0.10 (0.38) | -0.07 (0.25) | 86 | 47 |
| b | -0.52 (0.60) | -0.25 (0.81) | -0.75 (0.24) | 9 | 25 |
| c | -0.72 (0.41) | 1.07 (0.41) | -0.53 (0.25) | 28 | 116 |
| d | -0.95 (0.49) | 1.28 (0.41) | -0.58 (0.24) | 140 | 44 |



| | | | | | |
|---|---|---|---|---|---|
| e | -0.77 (1.39) | 1.49 (0.34) | -0.48 (0.16) | 128 | 38 |
| f | -0.97 (1.44) | -0.24 (0.36) | -0.17 (0.19) | 387 | 108 |
| g | -0.12 (0.52) | 0.23 (0.48) | -0.42 (0.15) | 204 | 88 |
| h | 0.97 (0.57) | -0.70 (0.41) | -0.28 (0.21) | 56 | 106 |
| i | 0.37 (0.67) | -0.24 (0.36) | -0.14 (0.15) | 87 | 230 |
| j | 0.24 (0.92) | 0.29 (0.40) | 0.04 (0.17) | 129 | 313 |
| k | 0.59 (0.79) | 0.11 (0.25) | -0.15 (0.20) | 57 | 207 |
| All intervals | -0.41 (1.19) | 0.26 (0.74) | -0.27 (0.27) | 1311 | 1322 |

**Table 6. Objective filtering diagnostics and accompanying threshold limits. † During standard operation conditions measurement duration is 10 minutes. * Additional diagnostics available after the FTIR upgrade in April 2013.**

| Diagnostic | Threshold filtering values |
|---|---|
| $H_2O$ (ppm) | X < 20 |
| $\Delta H_2O$ – Change in $H_2O$ between successive measurements (ppm)† | X <0.2 |
| Cell pressure (hPa) | 850 < X < 1105 |
| Cell pressure 1σ (hPa)* | X < 0.1 |
| $\Delta P$ – Change in cell pressure between successive measurements (hPa) | X < 1.4 |
| Cell temperature (°C) | 31.5 < X < 34.5 |
| Cell temperature 1σ (°C) | X < 0.02 |
| $\Delta T$ – Change cell temperature between successive measurements (°C) | X < 0.27 |
| Cell flow rate (Lmin$^{-1}$) | 0.43 < X < 0.65 |
| Cell flow rate 1σ (Lmin$^{-1}$) | X <0.015 |
| MALT retrieval root-mean-square error, for spectral regions 1,2,3 & 4 (RMSE) | X < 0.1, 0.03, 0.4, 0.01 |
| MALT retrieval spectral abscissa fitted shift, for spectral regions 1,2,3 & 4 (cm$^{-1}$) | X < 0.08, 0.12, 0.17, 0.075 |
| Time difference between sample and closest prior calibration | X < 8 days |

**Table 7. FTIR flask comparison results per species. FTIR flask dataset biases are listed with the 1σ standard deviation in brackets. Linear regression fitting parameters and uncertainties (bracketed) are listed in the middle columns. The final column has the Pearson correlation coefficient (r) of the fitted scatter plot data.**

| Species | Bias [ppb] | Simple linear regression | | r |
|---|---|---|---|---|
| | | gradient [ppb ppb-1] | intercept [ppb] | |
| $CH_4$ | -1.02 (2.61) | 0.97 (0.01) | 60.96 (24.02) | 0.99 |
| CO | -0.43 (1.60) | 1.03 (0.02) | -1.91 (1.15) | 0.99 |
| $N_2O$ | -0.01 (0.77) | 1.03 (0.05) | -8.93 (14.84) | 0.93 |


**Table 8. Trend analysis results (N=737) and bootstrap 1σ uncertainties (bracketed) for the period 2007-2017. The annual trend expressed as a percentage uses the timeseries mean dry mole fraction (1768.91 ppb, 55.16ppb and 324.29 ppb for CH₄, CO and N₂O respectively).**

| Species | Annual linear trend | | Peak-to-peak seasonal cycle amplitude [ppb] |
|---|---|---|---|
| | [ppb year⁻¹] | [% year⁻¹] | |
| CH₄ | 6.29 (0.23) | 0.36 (0.03) | 29.06 (0.86) |
| CO | -0.52 (0.29) | -0.94 (0.29) | 22.52 (0.71) |
| N₂O | 0.99 (0.01) | 0.31 (0.01) | 0.52 (0.04) |

**Figures and figure captions**

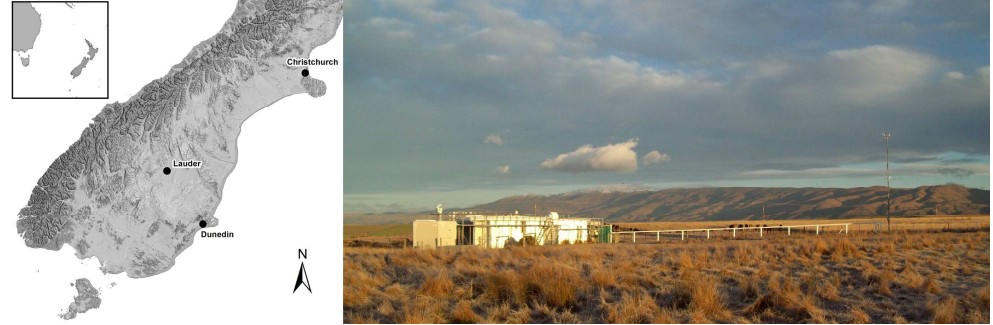

**Figure 1. (a) Location of Lauder, South Island, New Zealand. (b) A westward view of the in situ sampling mast and the building housing the in situ instrumentation.**

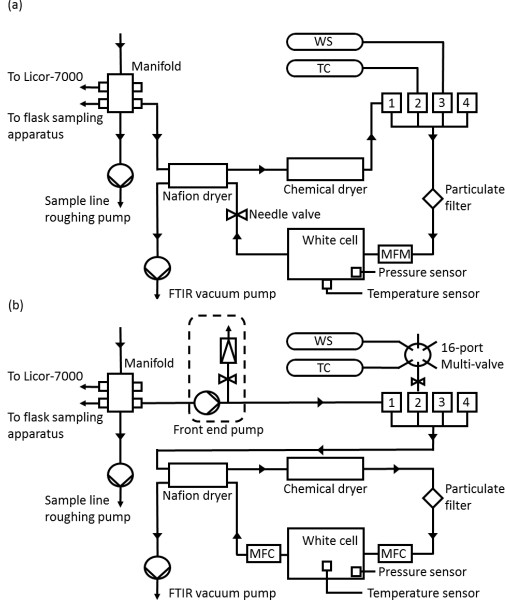





**Figure 2. (a) Simplified Lauder FTIR gas schematic prior to upgrades. WS = working standard, TC = target cylinder, MFM = mass flow meter. (b) Schematic of FTIR gas handling after the September 2013 upgrades. MFC = mass flow controller.**

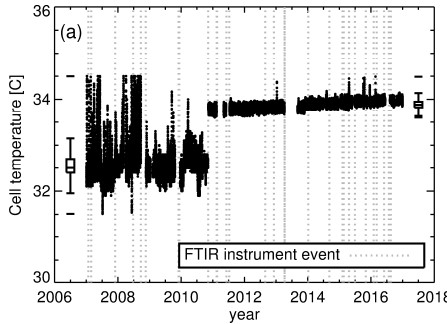

**Figure 3. Cell temperature measurements. From 2007 - September 2010 cell temperature measurements were made with an integrated circuit sensor attached to the outside of the cell. The invitro PT100 temperature measurements started in September 2010 and replaced with a Type-J thermocouple in April 2013 (measurements outside the range 31-35 °C were filtered out). Box plots provide a statistical summary prior and post LM335 temperature sensor change. Vertical grey dashed lines indicate an event in which changes to FTIR hardware, operating conditions or analysis were made (FTIR instrument events explained in Sect. 5.11).**

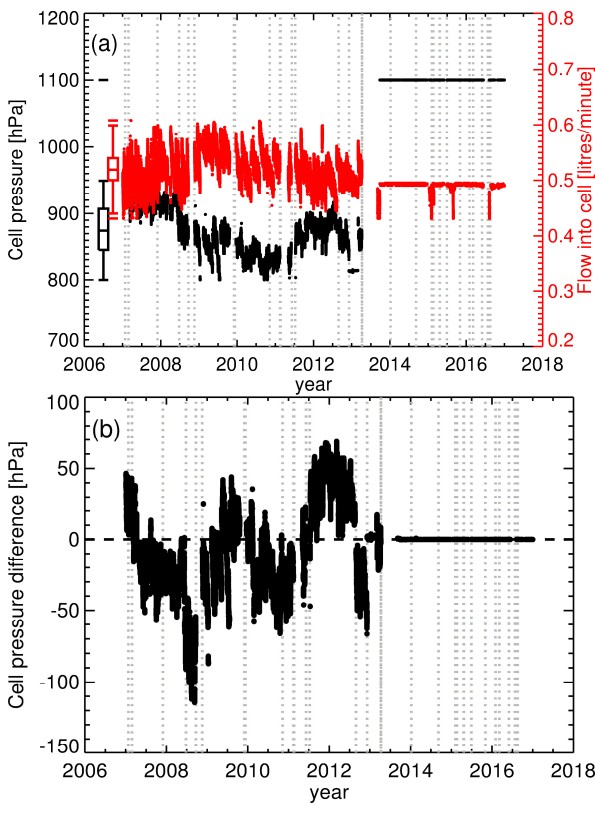



**Figure 4. (a) Cell pressure (black) and cell flow rate in flow mode (red) during air sample measurements. After the April 2013 upgrade the flow rate is set to 0.5 Lmin⁻¹ and cell pressure is set to 1100 hPa. The sudden drops in flow rate on three occasions (post upgrade) are due to MFC power supply faults. Data taken during such faults is filtered out. (b) Difference between air sample and WS cell pressure.**

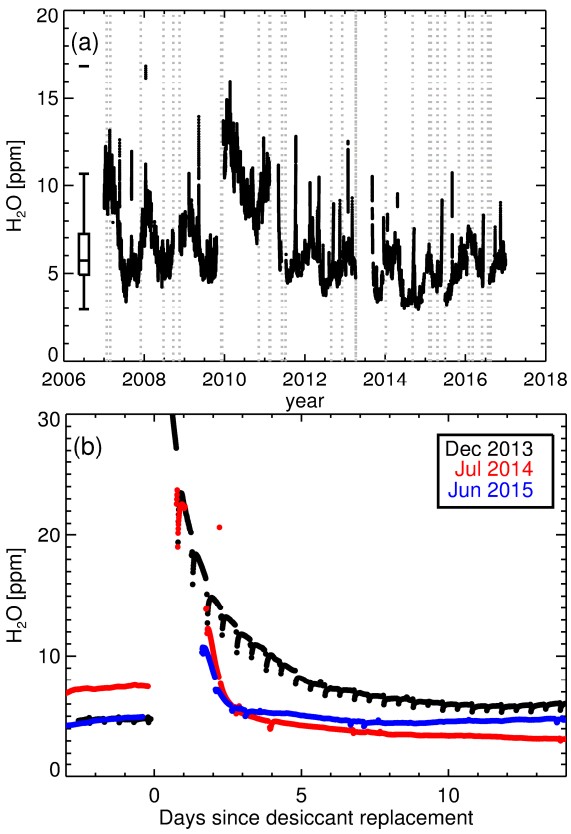

**Figure 5. (a) Retrieved H₂O during air sample measurements. The near-vertical spikes in H₂O relate to desiccant cartridge replenishment then subsequent drying out of the cell. All measurements with H₂O > 20 ppm are filtered out prior to calibration and analysis. The elevated level of H₂O (approx. 2 ppm) over 2010-2011 is unexplained. (b) Three examples of cell H₂O after desiccant cartridge replacement. The twice daily small reductions in H₂O in the Dec2013 data is due to daily calibration and target cylinder measurements.**



**Figure 6. Difference between the FTIR MKS 902 and external PTB110 pressure sensor prior to any calibration adjustments. Comparisons are conducted at a cell pressure of approx. 960 hPa (atmospheric pressure).**

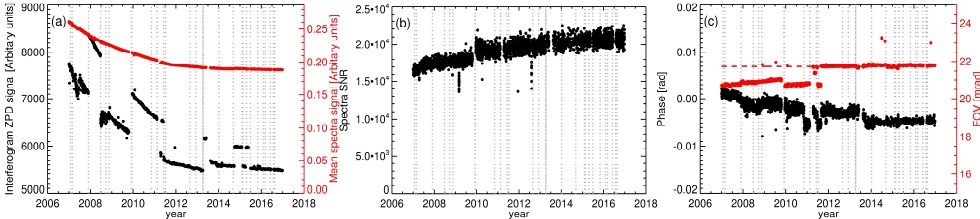

**Figure 7. (a) Interferogram ZPD signal and the mean signal level of the associated spectra calculated over the range 2450-2550 cm⁻¹. (b) Spectra SNR over the range 2450-2550 cm⁻¹. The 2450-2550 cm⁻¹ region was selected due to a lack of absorption features and is representative of the continuum (c) Fitted spectra phase and FOV.**

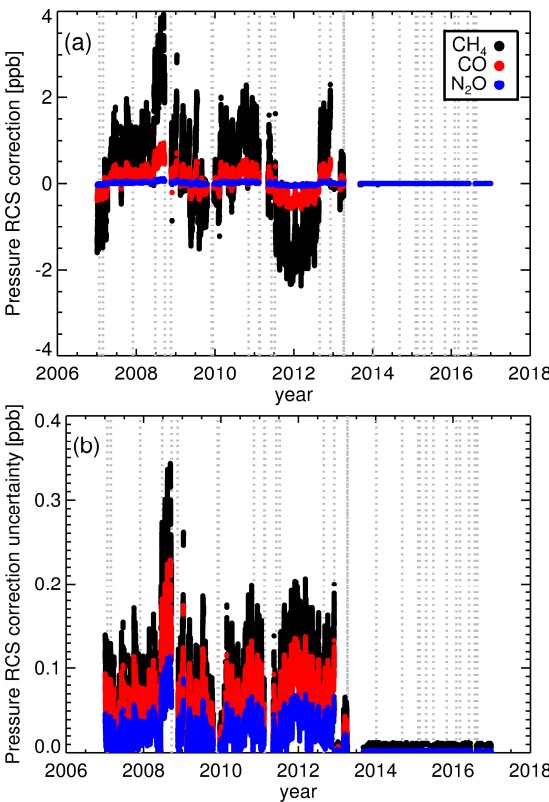

**Figure 8. (a) RCS$_p$ corrections applied to the calculated CH$_4$, CO and N$_2$O dry mole fractions and (b) associated correction uncertainties.**





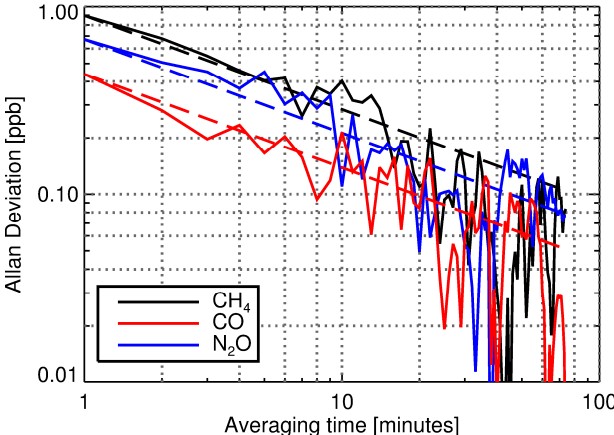

**Figure 9. CH₄, CO and N₂O Allan deviations calculated from the February 2015 repeatability experiment. The dataset consists of 170 consecutive 1-minute spectra acquired during measurement of the TC under standard operating conditions (cell pressure = 1100hPa, cell temperature = 33.85 °C, and a flow rate of 0.5 L min⁻¹). The dashed lines represent the Gaussian noise limited Allan deviation using the derived Allan deviation of the smallest temporal increment (1-minute) as the basis.**





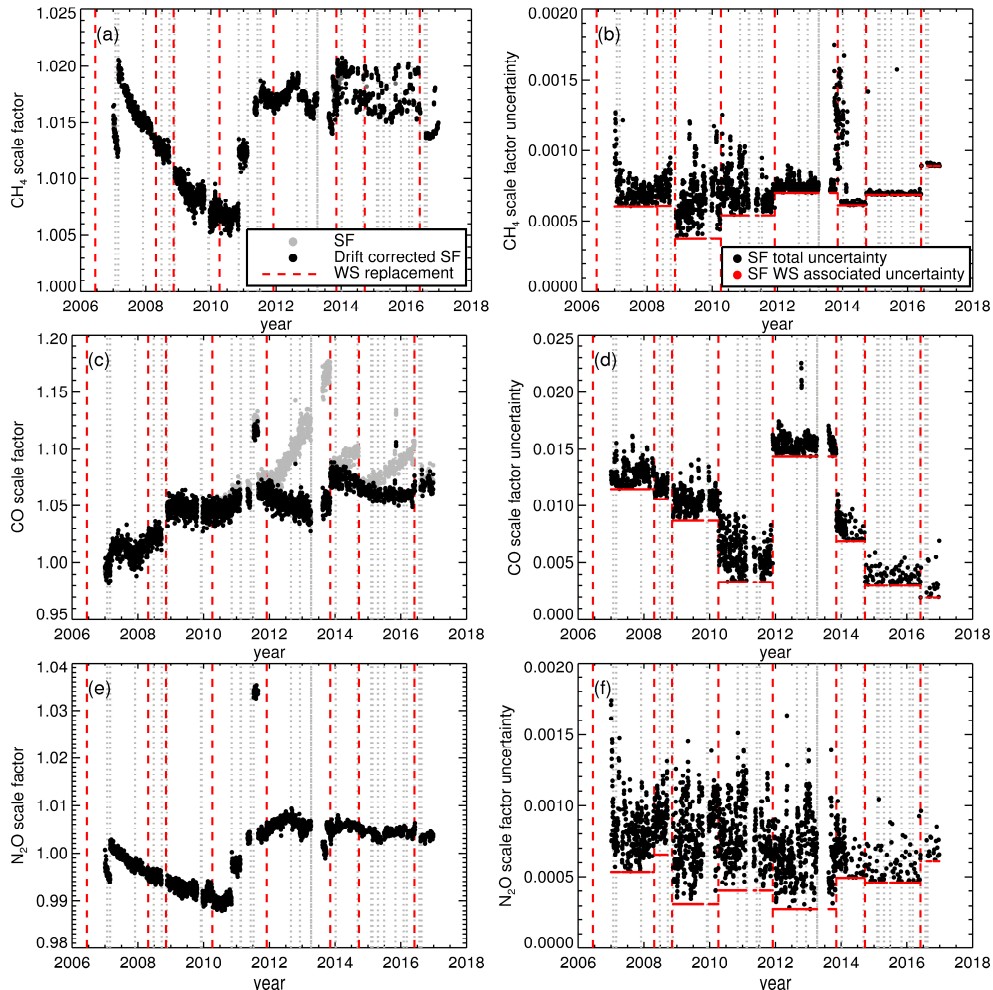

**Figure 10.** (a) CH₄ 7-day running mean scale factor (SF). Black data points are the drift corrected scale factors. Uncorrected scale factors are shown as grey data points. The vertical dashed red line indicates WS replacement and (b), CH₄ scale factor uncertainty. (c and d) same as (a and b) but for CO. (e and f) same as (a and b) but for N₂O.



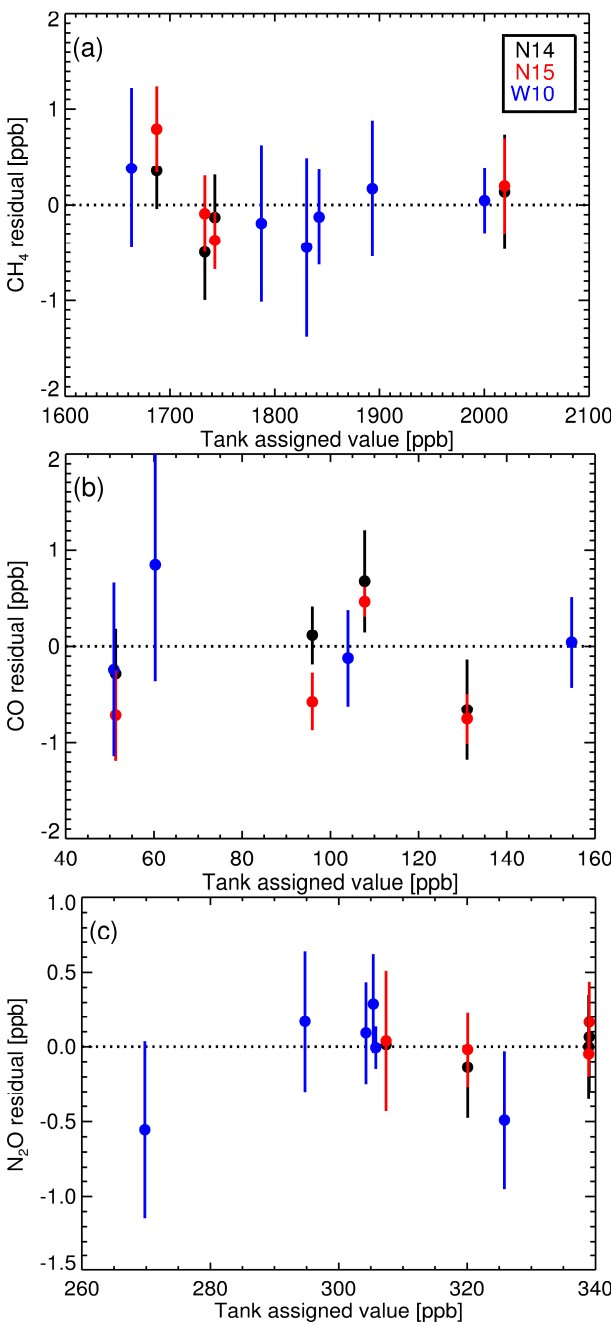

**Figure 11. Complete-IRF fit residuals (with 1σ uncertainty bars) from measurements of multi-tank suites N14, N15 and W10.**

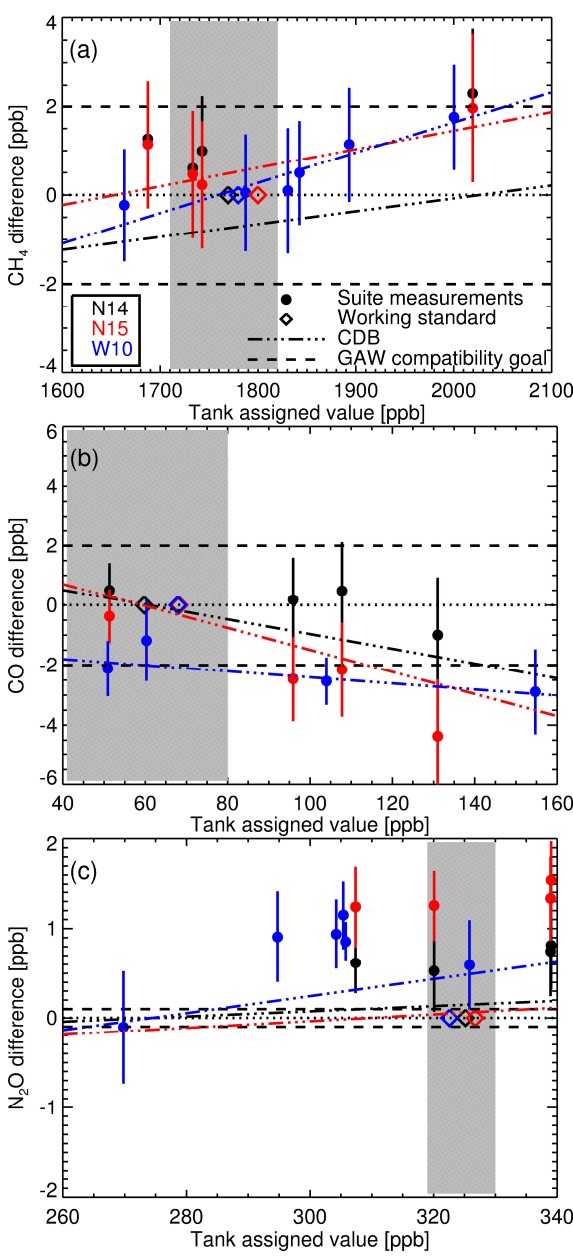

**Figure 12.** (a) The difference between calibrated **CH₄** measurements of the three multi tank suites (N14 black, N15 red and W10 blue) against assigned tank values with 1σ uncertainty bars. The coloured diamonds are the assigned WS dry mole fraction used to calibrate each respective set of suite measurements using the scale factor method. The coloured dash-dot-dot lines are the estimated concentration dependent biases (CDB) arising from applying the scale factor method, for each measurement suite. The grey shaded area indicates the typical baseline concentration range at Lauder. (b and c) the same as (a) but for CO and N₂O respectively.



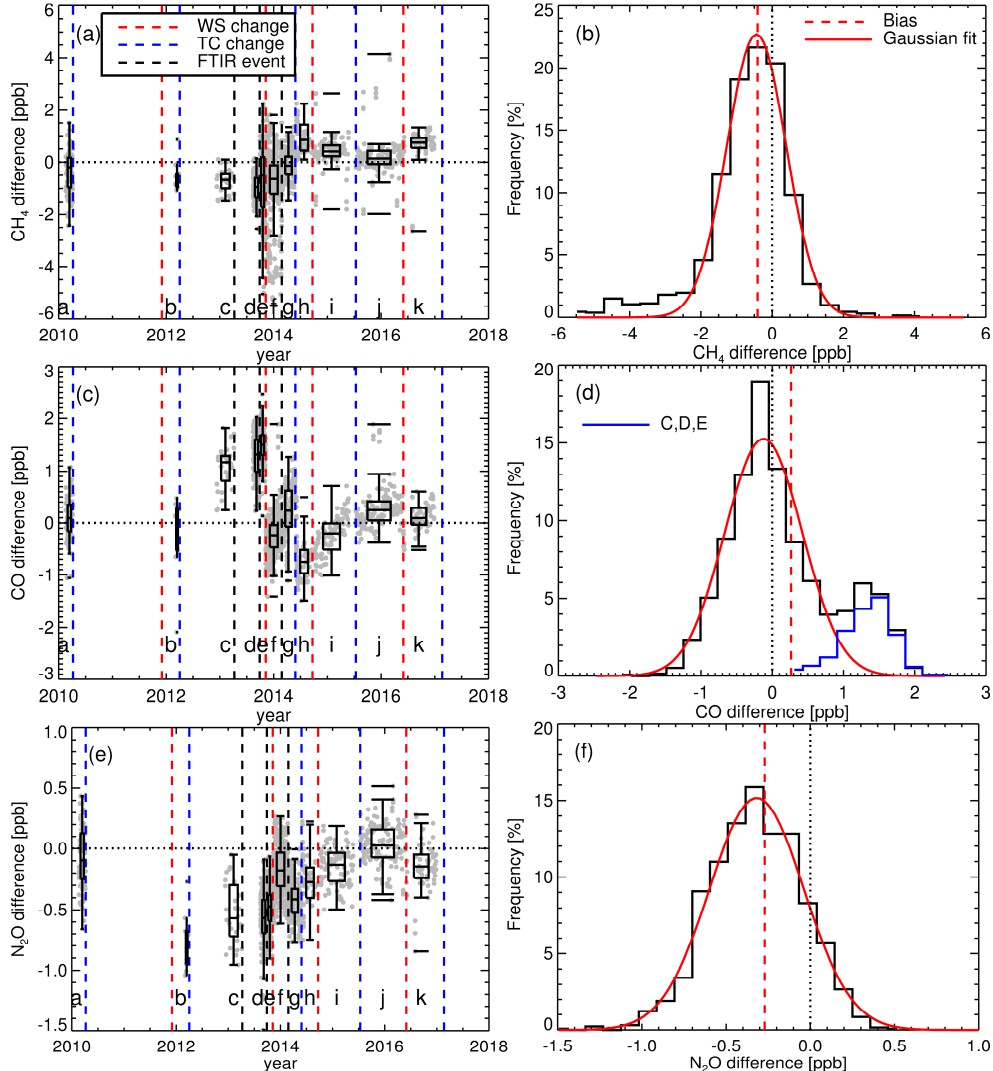

**Figure 13. (a) The grey data points show the difference between TC CH₄ measurements and that of the TC assigned values (FTIR - TC). The blue dashed vertical lines indicate TC change. The red dashed vertical lines indicate WS change. Black dashed vertical lines indicate a significant instrument event. The intervals between changes have alphameric labels. Box plots display interval summary statistics. (b) Histogram of FTIR-TC flask differences over all intervals. The dashed vertical red line is the mean difference (bias). The red line is a Gaussian fit to the histogram to illustrate the deviation of the differences from that of a theoretical random Gaussian statistical distribution based upon the given dataset. (c and d) the same as (a and b) but for CO respectively. In (d), the additional blue histogram relates to intervals C, D and E. (e and f) the same as (a and b) but for N₂O respectively**



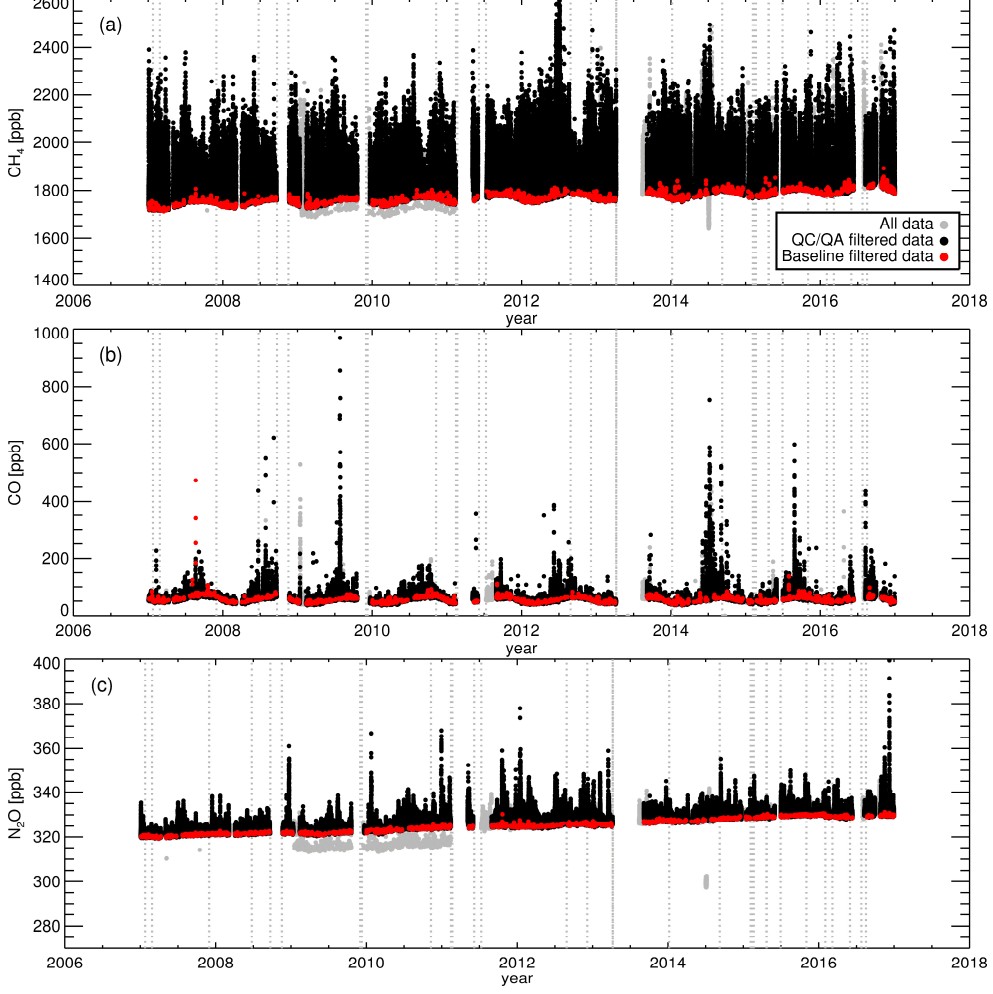

**Figure 14. (a) Calibrated time series of CH₄, (b) CO and (c) N₂O.**





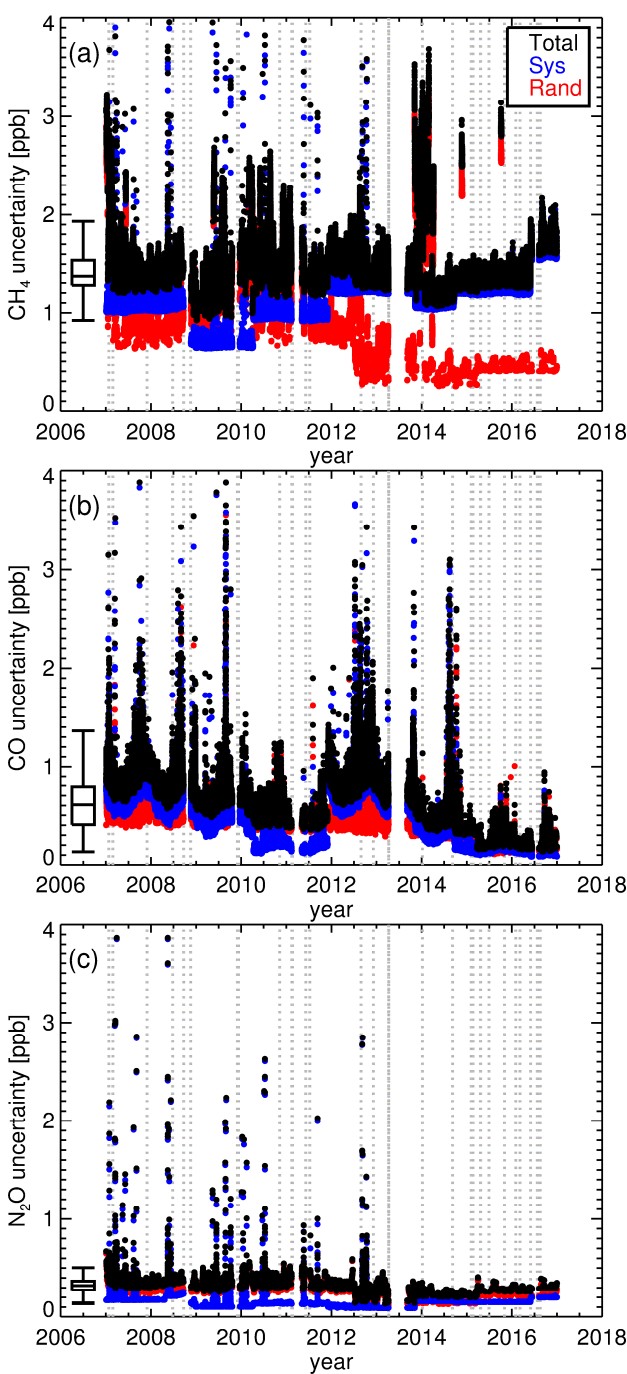

**Figure 15.** (a) CH₄ measurement uncertainties: total, systematic (Sys) and random (Rand). A box plot statistical summary for total uncertainty is overlaid. (b and c) the same as (a) but for CO and N₂O respectively.





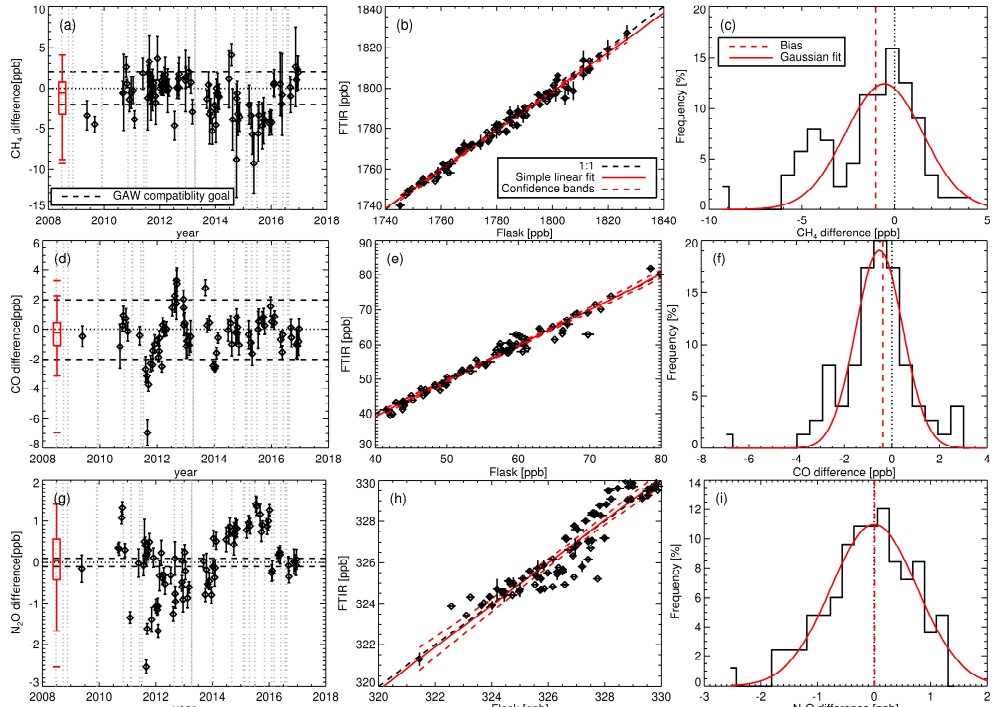

**Figure 16. (a) CH$_4$ FTIR flask comparison. A time series of differences between FTIR and flask measurements (FTIR minus flask). Error bars are the uncertainty in the flask measurements added in quadrature with the FTIR 1-hour variability. The horizontal dashed line is the GAW recommended compatibility goal. A box plot statistical summary of the FTIR flask differences is overlaid in red. (b) CH$_4$ FTIR flask correlation plot. The simple linear regression line is over plotted in red. The red dashed lines are the Working–Hotelling 90% confidence bands. For reference, the 1:1 correlation line is indicated the black dashed line. (c) Histogram of FTIR flask differences. The dashed vertical red line is the mean difference (bias). The red line is a Gaussian fit to the histogram to illustrate the deviation of the differences from that of a theoretical random Gaussian statistical distribution based upon the given dataset. (d-f), same as (a-c) but for CO. (g-i) same as (a-c) but for N$_2$O.**



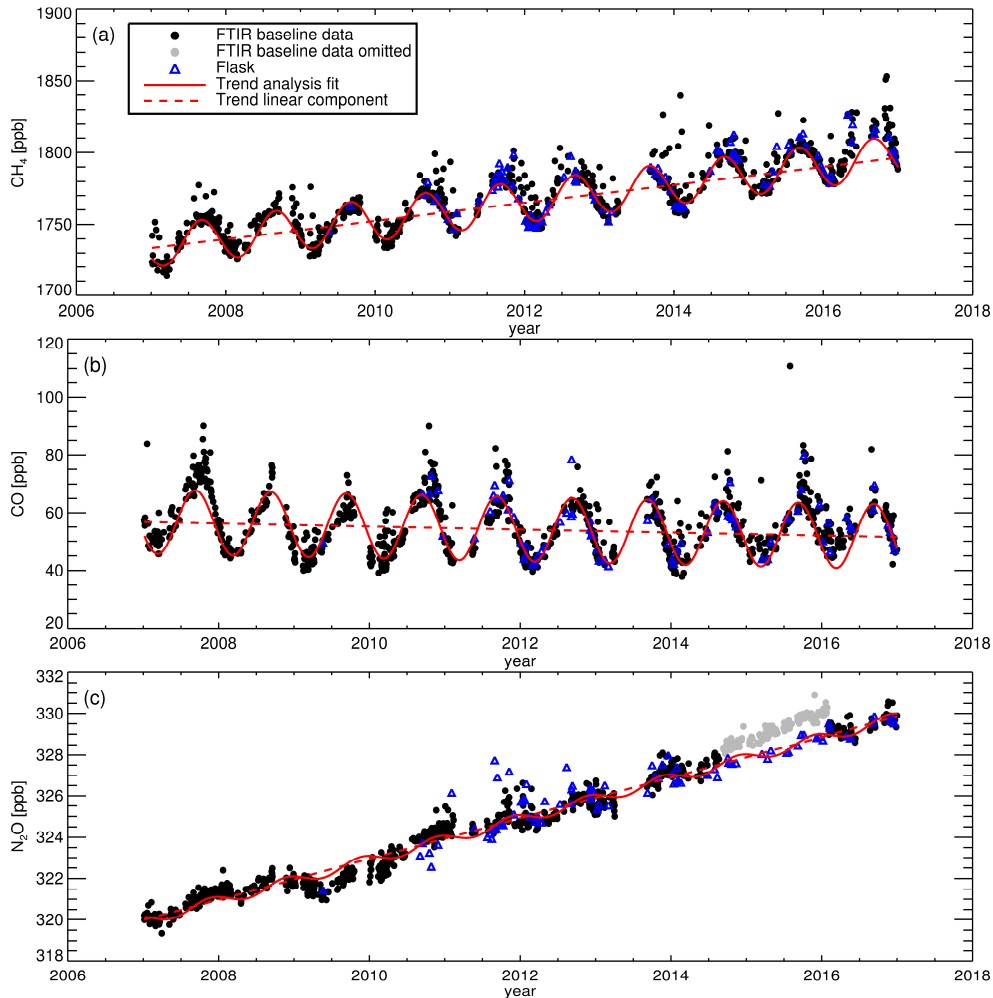

Figure 17. (a) Baseline CH₄ , (b) CO  and (c) N₂O FTIR measurements and flask samples. FTIR trend analysis fit and
the trend analysis linear fit component are over plotted.



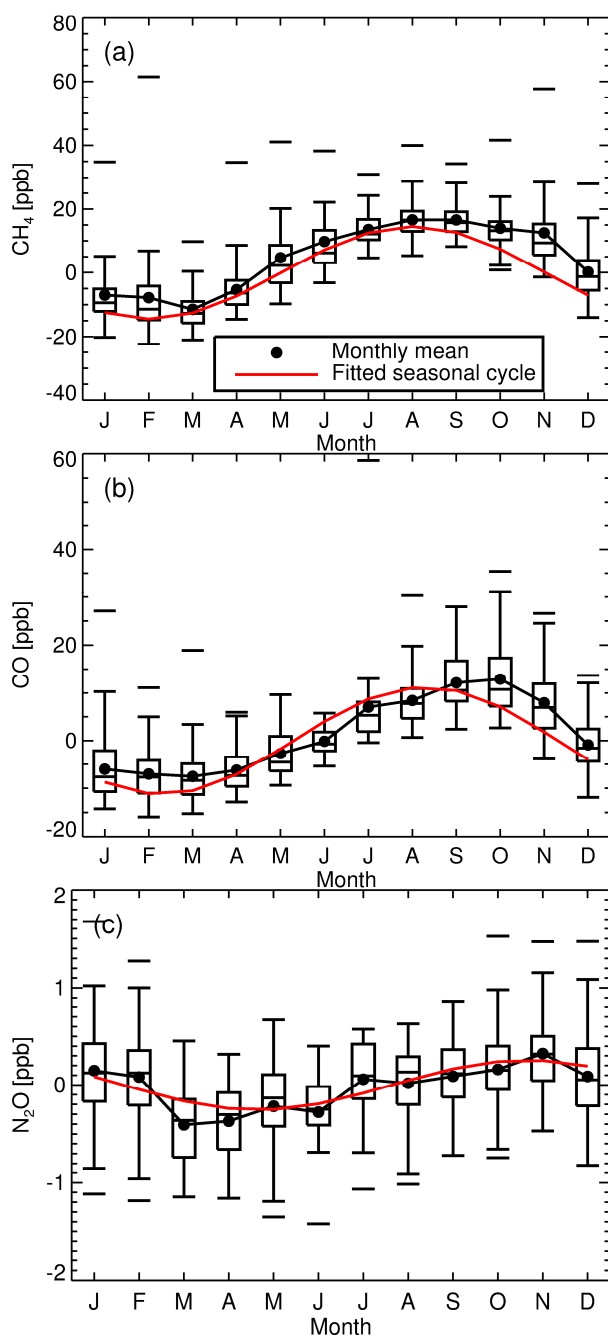

**Figure 18. Box plot statistical summaries of detrended monthly baseline measurements of CH₄ (a), CO (b) and N₂O (c) over the period 2007-2017. The fitted seasonal cycle (1 Fourier pair) is overlaid as a solid red line.**




## Appendix A

**An updated retrieval strategy for CO and N₂O**

### A1. Reduction of CO and N₂O residual cross sensitivity to $^{12}CO_2$.

In the original spectral analysis strategy employed at Lauder three broad spectral regions were analysed, R1-R3
in Table A1. H13 found a significant non-linear cross sensitivity between CO and N₂O to $^{12}CO_2$. To minimize these cross sensitives an additional spectral region was added; R4: 2097–2242 cm$^{-1}$. Spectral region absorption examples are found in G12 Fig. 3. Experiments show that R4 CO and N₂O retrievals have $^{12}CO_2$ linear cross sensitivities of the order -0.002ppb ppm$^{-1}$ and 0.0001 ppb ppm$^{-1}$ respectively, which are relatively inconsequential. There was no substantial change in CO and N₂O precision. An additional benefit is a reduction
in the MALT CO retrieval sensitivity to temperature and pressure measurement errors (listed in Table A2). For N₂O, pressure sensitivity in R4 retrievals is similar to that in R1 along with an (undesirable) increase in temperature sensitivity. The CO and N₂O retrieval sensitivity to $CO_2$ forward model error (dX/dCO₂) is also listed. R4 CO and N₂O retrievals are far less susceptible to a forward model $CO_2$ error, this is more theoretical than practical as $CO_2$ is also retrieved but provides an indication of the need to fit $CO_2$ correctly and indicates an
overall robustness of the retrieval strategy. R1 retrievals are still required for $^{13}C-CO_2$, in which CO and N₂O are regarded as interfering species.

| Region | Spectral range (cm$^{-1}$) | Retrieved target species | Retrieved interfering species |
|--------|------------------------|--------------------------|-------------------------------|
| R1 | 2150–2320 | $^{13}CO_2$, $^{12}CO_2$, CO and N₂O | $H_2O$, $^{12}C^{18}O^{16}O$ |
| R2 | 3001–3150 | CH₄ | $H_2O$ |
| R3 | 3520–3775 | $CO_2$ | $H_2O$ |
| R4 | 2097–2242 | CO, N₂O | $CO_2$, $H_2O$ |

**Table A1. MALT retrieval spectral regions and retrieved species within each region.**

| | CO | | N₂O | |
|---|---|---|---|---|
| | R1 | R4 | R1 | R4 |
| dX/dT [ppb C$^{-1}$] | -4.43 | 0.17 | 1.65 | 2.16 |
| dX/dP [ppb hPa$^{-1}$] | 0.27 | -0.07 | -0.35 | -0.33 |
| dX/dCO₂ [ppb ppm$^{-1}$] | -3.30 | 0.02 | -0.24 | -0.02 |

**Table A2. MALT CO and N₂O retrieval sensitivity to pressure (dX/dP) and temperature (dX/dT) measurement errors, in the two spectral regions R1 and R4. The CO and N2O retrieval sensitivity to $CO_2$ forward model error (dX/dCO₂) is also listed. These were derived in a theoretical MALT study using perturbed pressure (1hPa, 10hPa),**
**temperature (0.1 °C, 1.0 °C) and CO₂ (1ppm, 10ppm) using a typical air sample composition (CO₂: 390ppm, CH₄: 1800ppb, N₂O: 320ppb and CO: 50ppb) in standard conditions (P = 972hPa, T = 32°C).**

The entire Lauder FTIR CO and N₂O dataset was reanalysed with the R4 CO and N₂O retrieval strategy and is now part of routine MALT analysis. Comparing MALT CO retrievals from spectral regions R1 and R4 over a three-month period gives a bias of 3.6 ± 0.38 ppb (R4 higher). Over the same period there is a bias of -1.6 ±


0.2ppb between MALT $N_2O$ R1 and R4 retrievals. Such biases are not significant as they are cancelled out during the calibration process.

**A2 Elimination of $N_2O$ non-linear pressure residual cross sensitivity**

A serendipitous consequence of adopting the new R4 region for $N_2O$ spectral analysis is the elimination of significant $N_2O$ $RCS_p$ non-linearity observed in R1 spectral retrievals. R1 $N_2O$ $RCS_p$ parametrisation required a $2^{nd}$ order polynomial fit. The $N_2O$ $RCS_p$ derived from R4 spectral analysis can be approximated as a linear function. An example of $N_2O$ $RCS_p$ calculated using spectral regions R1 and R4 are illustrated in Fig. A1. The difference in the retrieved dry mole fractions is not of concern as calibration procedures will determine the absolute accuracy.

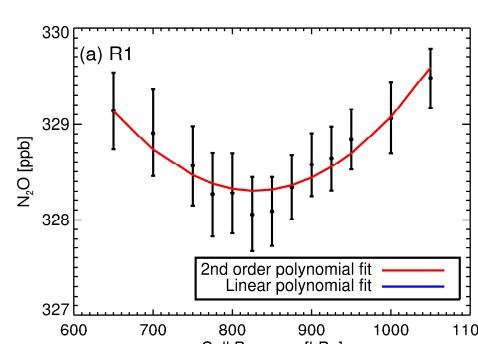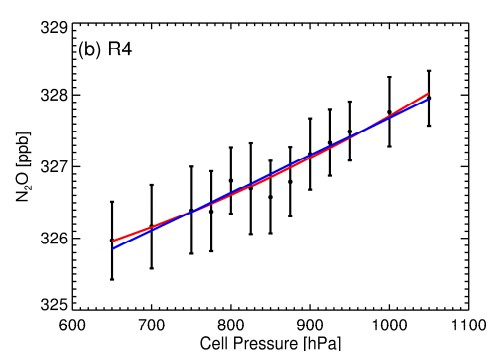

Figure A1. Retrieved $N_2O$ dry mole fractions as a function of cell pressure from tests conducted in December 2013. (a) R1-$N_2O$ spectral analysis (with 1σ uncertainty bars). (b) Same as (a) but for R4-$N_2O$ spectral analysis.

**Appendix B**

**Background spectrum water vapour removal**

Spectral analysis is performed on transmission spectra, not the actual collected raw sample spectra. Sample transmission spectra are generated by the ratio of the measured spectra to that of a reference background spectrum. Reference background spectra are collected under the same experimental set up as that of measurements but taken when the cell is evacuated. Using transmission spectra rather than raw sample spectra eliminates instrument artefacts such as continuum level curvature and the spectrometer's spectral response.

It was initially observed that retrieved species dry mole fractions were dependent on the background spectrum. When a transmission spectrum is calculated there is imperfect spectral cancellation of residual water absorption lines between raw sample spectra and collected background spectra. Species absorptions of interest (e.g. $CO_2$, $CH_4$, CO and $N_2O$) which are heavily overlapped by water vapour absorptions are most effected. This primarily effects the retrieval of $CO_2$ in the broad spectral region 3520–3775 cm$^{-1}$ whilst retrieved $CH_4$, $N_2O$ and CO are mostly unaffected. Water vapour absorption does not 'ratio out' simply or linearly when calculating a transmission spectrum, for two reasons, first, because the sample and background spectra are recorded and apodised by the FTIR to 1 cm$^{-1}$ before being divided to calculate transmission spectra, in which cancellation is not complete (this is a consequence of the breakdown of Beer's law at low resolution (Griffith, 1996)). Second,



the background water vapour spectrum has two components, water vapour at approx. 1hPa (evacuated cell pressure) and residual water vapour at atmospheric pressure (approx. 1300 hPa) in the IRcube transfer optics compartment which is purged with dry nitrogen. Since the low-pressure spectral absorption lines are narrower, the spectral line shapes are not identical and do not provide a clean subtraction of water vapour in the sample

spectrum (approx. 1100 hPa). The result is that the transmission spectrum calculated has three water vapour components, of which there is not full cancellation. To account for this behaviour one method is to remove the water vapour absorptions from the background spectrum.

A water-absorption free background spectrum is constructed by fitting a small region of the measured

background spectrum with a 2-layer MALT model, one layer at 1 hPa the other at 1300 hPa, to retrieve the water vapour amount in the background spectrum. The concentrations and ILS parameters from this fit are used as input to MALT, in simulation mode, to simulate the transmission spectrum of water vapour in two layers at the levels in the selected background spectrum. The measured background spectrum is then divided by the simulated water transmission spectrum.  The result is a water-absorption free background spectrum. This desiccated (or so

called stripped) simulated background spectrum is used when constructing transmission spectra from air samples. The retrieved water vapour from sample transmission spectra is now only that attributed to the water vapour in the sample spectra, and the fits are generally good with very small residuals. As illustrated in Fig. B1, the stripped background in the R2 and R4 spectral regions are unaffected by the removal of the water absorption features. $CO_2$ retrieved in region R3 along with $^{12}CO_2$ and $^{13}CO_2$ in region R1 are the most affected.

A similar stripping procedure is used to remove residual $CO_2$ absorption in the background spectrum due to incomplete purging of the IRcube and evacuation of the cell.

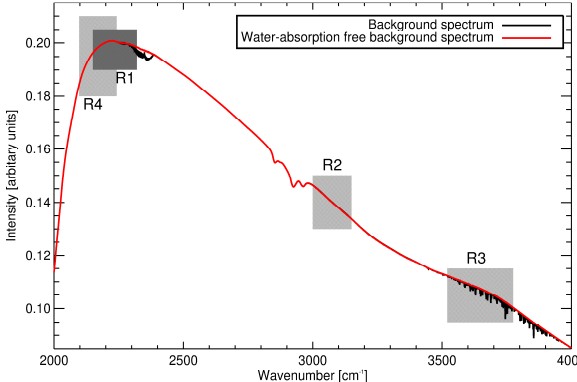

**Figure B1. A typical background spectrum taken on 8 August 2014 (cell pressure of 1.6 hPa) and corresponding background spectrum with water absorption spectral features removed. MALT spectral fit regions are shaded in grey.**

Experiments were conducted to investigate and quantify the effect of using stripped background spectra in $CO_2$ retrievals. Sample spectra were taken of a single ambient air tank. The tank air was pre-conditioned with variable



amounts of water vapour (10-250 ppm) prior to delivery to the FTIR. Four background spectra were also taken, with differing amount of water vapour (spanning 0.01 - 0.47 ppm). For each background spectrum, a simulated stripped background spectrum was made. Each sample spectra were then ratioed to these eight background spectra to make transmission spectra. The transmission spectra were then analysed with MALT in the standard way.

As illustrated in Fig. B2 there is a $CO_2$ concentration dependence on both the amount of water vapour in the sample and background spectra water vapour content when using unstripped backgrounds (red data points). For stripped backgrounds, all four $CO_2$ retrievals agree to within 0.5 ppm (for CO, $N_2O$ and $CH_4$ the difference was 10 times less than instrument precision). The dependence of $CO_2$ on sample water vapour is reduced by more than a factor of ten relative to the wet backgrounds. These results indicate that it is inaccurate fitting of the composite water vapour spectrum when using unstripped backgrounds that leads to the sample water vapour dependence of $CO_2$ retrieval. With only sample water vapour to be fitted, MALT can do a good fit and there is little cross-sensitivity. The differences between the four stripped backgrounds reflect small changes in the overall response of the FTIR spectrometer, with the probable cause being temperature stability.

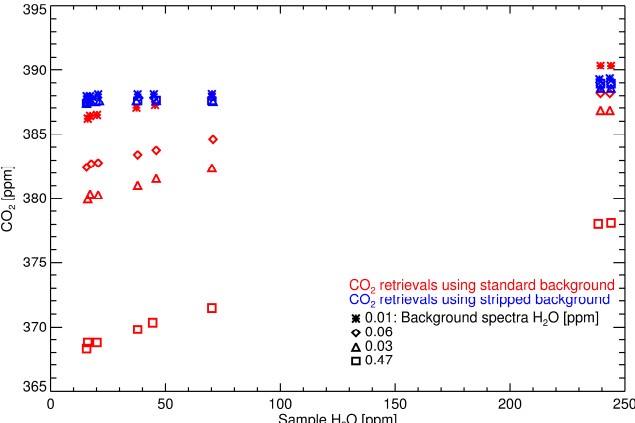

**Figure B2. Retrievals of $CO_2$ dry mole fractions from a standard cylinder that has been preconditioned with water vapour of differing amounts using unaltered background spectra (red) and stripped background spectra (blue). The legend displays the amount of retrieved water vapour in the four background spectra. All background spectra were taken with the cell evacuated to approx. 1 hPa.**

**Appendix C**

**A customized scale transfer reference tank suite**

A bespoke FTIR scale transfer reference gas four tank suite (referred to as the Aniwaniwa suite) was designed by NIWA and prepared at the National Oceanic and Atmospheric Administration Earth System Research Laboratory Global Monitoring Division (NOAA ESRL GMD, Kitzis, 2017). NOAA ESRL GMD acts as the WMO-GAW Central Calibration Laboratory for $CO_2$, $CH_4$, $N_2O$ and CO. The suite has a customized trace gas composition matrix consisting of prescribed $CO_2$, $CH_4$, $N_2O$ and CO dry mole fractions calibrated on the respective WMO reference scale. The prescribed dry mole fractions span the typical sample air trace gas dry





mole fractions measured at Lauder. The $\delta^{13}$C-CO$_2$ isotopic composition of the Aniwaniwa suite were assigned at NIWA-Gaslab employing GC isotope ratio mass spectrometry using VPDB scale transfer reference gases.

The composition matrix (listed in Table C1) was designed to minimize species cross sensitivity/covariance in the
MALT retrieval algorithm. Preference for species concentration orthogonality is given to species retrieved in the same spectral region (for example CO and N$_2$O). There is insignificant covariance between species retrievals in differing spectral region. The MALT retrieval code performs spectral fitting in four independent spectral regions (listed in appendix A). The original retrieval strategy only used three spectral regions: R1, R2 and R3. This was expanded to four to minimize N$_2$O residual cross sensitivity to CO$_2$. The Aniwaniwa suite was constructed prior
to the retrieval strategy update change, hence the suite composition matrix is based around minimizing species concentration correlation for each tank based on retrievals in spectral regions R1, R2 and R3.  This is not of major concern as spectral region R4 has a large overlap with R1.

    Tanks CB09978 and CB10202 have the same N$_2$O dry mole fractions within uncertainty limits (0.08pbb
difference). Tanks CB09978 and CB10248 also have similar CH$_4$ dry mole fractions (9.58ppb difference). For these species, the effective suite tank span reduces from 4 to 3 tanks, but still important as overall tank composition differs. NOAA ESRL GMD keep a full audit history of tank preparation and scale propagation. Tank assignment changes and/or reference scale changes are accessed via the public accessible site: https://www.esrl.noaa.gov/gmd/ccl/refgas.html. All four tanks were delivered with an approx. pressure of 2000
psig.

| Tank ID | CH$_4$ [ppb] WMOx2004A R2 | CO [ppb] WMOx2014 R4 | N$_2$O [ppb] WMO2006A R4 | CO$_2$ [ppm] WMOx2007 R3 | $\delta^{13}$C-CO$_2$ (‰) VPDB R1 |
|---------|---------|---------|---------|---------|---------|
| CB09978 | 1733.24 (0.13) | 95.90 (0.13) | 339.02 (0.11) | 412.70 (0.01) | -8.774 (0.005) |
| CB10005 | 1687.32 (0.27) | 131.01 (0.03) | 320.08 (0.10) | 398.51 (0.03) | -8.662 (0.004) |
| CB10248 | 1742.82 (0.22) | 51.32 (0.28) | 307.38 (0.13) | 457.68 (0.06) | -8.804 (0.005) |
| CB10202 | 2019.30 (0.13) | 107.77 (0.20) | 338.94 (0.15) | 380.42 (0.01) | - |

**Table C1. Aniwaniwa suite composition with assignment uncertainty bracketed (1σ). $\delta^{13}$C-CO$_2$ was not assigned at NOAA ESRL GMD, but measured at NIWA-Gaslab. The current WMO reference scales are given along with the**
**spectral analysis region retrievals are performed in.**

**Appendix D**

**Defining baseline conditions**

We define baseline measurements as those taken in conditions that are representative of a well-mixed boundary
layer devoid of any local source emissions.  A simple physical based approach is taken in defining what baseline conditions are at Lauder, this is when the windspeed is greater than 5ms$^{-1}$, between 1500-1600 NZST, and there are more than five samples taken within this hour. This last criterion allows baseline measurement variability to be quantified. We also found that wind direction did not need to be considered for baseline filtering. Such



filtering is applied to all three species. Due to the lack of consistent local emission sources the current baseline definition is sufficient for our needs. A more sophisticated approach in defining baseline conditions is possible (e.g. Stephens et al. 2013, Yuan et al. 2018). Identifying local emission spikes using methods like that proposed by El Yazidi et al. (2018) could also be used.

Figure D1 shows the $CH_4$ hourly standard deviation as a function of wind speed and time of day. From these figures, we see reduced $CH_4$ variability with higher wind speeds with wind speed greatest (but also with highest variability) in the mid-afternoon through to early evening. Greater wind speeds produce more regional mixing creating a more homogenous atmosphere. The $CH_4$ hourly standard deviation diurnal cycle is at a minimum in

10   the early to mid-afternoon. Such local afternoon minima are also seen in CO, $N_2O$ (not shown) and $CO_2$ (Steinkamp et al., 2017). The large variability at night time is due to the formation of a nocturnal boundary layer during certain meteorological conditions.

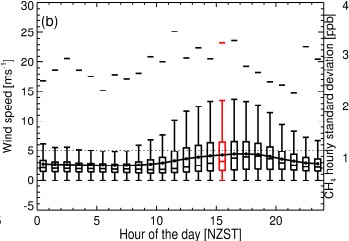
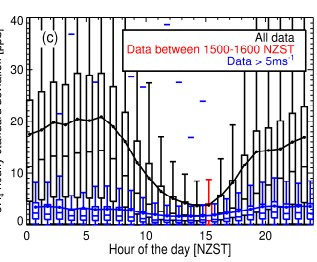

15   **Figure D1. (a) $CH_4$ hourly standard deviation (minimum of five samples) as a function of wind speed, and data filtered by time of day (red) and full baseline criteria (blue). (b) Box plot statistical summary of hourly wind speed. (c) Box plot statistical summary of $CH_4$ hourly standard deviation. Note, some of the box plots upper outliers are truncated.**