# Peer review of "A decade of $CH_4$ , CO and $N_2O$ in situ measurements at Lauder, New Zealand: assessing the long-term performance of a Fourier transform infra-red trace gas and isotope analyser."

_Atmospheric Measurement Techniques, 2018_

## Referee Comment (RC1) · Anonymous Referee #1 · 9 Nov 2018

This paper describes the history and technical aspects of an in situ analyser that has been running in Lauder, NZ since January 2007. The spectrometer measures a variety of trace gas species, but this paper focuses on methane, carbon monoxide, and nitrous oxide. This paper describes a high quality, unique dataset that promises to be very valuable to the scientific community.

General comments:

The paper is well written, although it is quite long and reads much like a technical report

or detailed owners manual. The most interesting parts of the paper (in my opinion) are sections 1, 5.12, 6, and 7. I was unable to determine why some sections of the paper were put into appendices whereas other sections were not. I recommend that the authors consider putting more of section 5 into appendices, and in the main body state only information that is required to understand the time series analysis.

Specific comments:

P2L12: You may want to motivate your work by reminding the reader that there are few emissions in the SH, so these SH mid-latitude measurements are crucial for pinning down the true background values.

Technical comments:

P14L17: Did you also assess the modulation efficiency of the FTIR, along with the phase and FOV?

P15L25: I would have liked to see more of these linear regression curves. You show one in the appendix, so either refer to that figure or plot a few more here.

P16L8: You have not yet defined QC/QA.

P16L23: How do you determine RCSp?

P18L3: retrieved dry mole fractions *to* that of the assigned

P18L5: Do you mean to refer to Eqn (2)?

P19L28: A step change is an indication *of* an acute incident

P22L27: Suggestion: "Our approach is to take regular measurements..."

P26L16: This approach would *be* need*ed* when comparing...

Figure captions: Please make the figure captions self-explanatory. For example, Fig 10 shows scaling factors, but does not discuss what is being scaled.

Fig 17: I had trouble seeing the blue triangles. Could you make them bigger?

[Figure]

---

## Referee Comment (RC2) · Anonymous Referee #2 · 18 Nov 2018

Smale et al. describe and document in this paper a ten year time-series of continuous Greenhouse Gas mole fractions measured using a FTIR analyser at Lauder, New Zealand. They describe the improvements introduced to the measurement setup and the instrument and evaluate how these affected the measurement precision and accuracy. Unfortunately they do not describe the results for CO2 and 13CO2 in this paper and focus only on CH4, CO and N2O.

General comments:

I concur with reviewer #1 that the paper is generally well written but way too long. Many of the detailed descriptions could be abbreviated with at least 50% or be transferred into the appendices (e.g. sections 5.4 and 5.5).

Although the paper claims that this is the longest time series from this kind of instrument to date, this could be taken with a grain of salt, one could argue that the actual homogeneous time series only starts after the many changes in setup that took place up until Feb 2014.

However, the careful evaluation of measurement biases and precision as a function of time as performed here are a significant improvement over just providing the mole fraction time-series, and should be recommended good practice for all published GAW in-situ observations.

Specific comments:

The paper refers in the abstract to the compatibility goals as set by WMO GAW for greenhouse gas observations and compares the most recent results after all improvements and fine tuning to these by looking at the comparison with analyses of flask samples. Although the comparison with flask samples is a useful and common measure for quality assurance it is not the most authoritative measure. As in section 6 the rejection criterium for the duplo analysis of the flasks for N2O at the NIWA GC system is set to 0.5 ppb, I assume that the GC analysis reproducibility is about half of this and thus we cannot expect that the comparison between flask and FTIR measurements will be conclusive and be better than this 0.3 ppb. A better comparison would be to compare with in-situ continuous observations with different techniques such as CRDS or QCL or results of one of the round-robin exercises from the GAW CCL.

Technical comments:

P8L8: for the PT100 RTD one should specify the tolerance class, the resolution of the transmitter is not that relevant as long as it is order of magnitude better than the

tolerance class value. From the value specified in P8L16 one might guess the tolerance class is F 0.1.

P8L19: A thermocouple will show significant more short term and long term drift than any PT100 so the reason for this change is questionable. There also very thin, fast response time, PT100 RTDs.

P10L35:P11L8: There will be a small residual of sample air (1/200*1/870) left in the WS and TC air samples, is this corrected for in the analyses by using the mole fractions determined in the previous sample?

P10L35:P11L8: Why were the WS and TC measurements not performed in duplo or triplet? This would allow to detect offsets due to differences between flow and static mode especially for the first filling due to for example differences in water vapor content, this was recognized by the authors as since Feb 2014 the first calibration result is always skipped (P11L24). How big was the effect there?

P19L28: indication an -> indication of an

P19L34: approx. -> approximate

P30L25: The link given to the data will become obsolete after November 2018, as this website will be shutdown by JMA. The new WDCGG site is: https://gaw.kishou.go.jp/. It would be good to have the total uncertainty and bias estimates as in figure 15 also available together with the mole fraction time series in the same file or as a separate datafile.

Figure 16 there seems to be a cluster of obs for N2O where flask measurements are higher than the FTIR. It would be useful to see if the lower ring of dots below the 1:1 line between flask 325-328 ppb and FTIR 325-327 is a cluster connected in time that could be removed due to a problem in either GC or FTIR obs.

---

## Author Comment (AC1) · 13 Dec 2018

Reply to anonymous referee #1

We gratefully appreciate the time taken for the referee to read and evaluate our manuscript. We thank her/him for the helpful comments and suggestions to clarify issues and to improve the content, readability and presentation of the manuscript. Below we address each question, suggestion, correction or criticism individually. Referees' comments are shown in blue. Responses are in regular font. Quotes from the manuscript are in quotation marks, with altered manuscript wording given in bold type.

This paper describes the history and technical aspects of an in-situ analyser that has been running in Lauder, NZ since January 2007. The spectrometer measures a variety of trace gas species, but this paper focuses on methane, carbon monoxide, and nitrous oxide. This paper describes a high quality, unique dataset that promises to be very valuable to the scientific community.

General comments:

The paper is well written, although it is quite long and reads much like a technical report or detailed owner's manual. The most interesting parts of the paper (in my opinion) are sections 1, 5.12, 6, and 7.

Reduction of manuscript length was also recommended by referee 2. We are glad to hear the referee enjoyed sections 1, 5.12, 6 and 7. The content is technical and detailed by the nature and focus of the manuscript (and stated as such in the last paragraph of the introduction). We are looking at the long term technical performance of the FTIR analyser system (both good and bad points), which no one has done before. We would like to think that the technical details presented would assist a research group that was thinking of setting up a long-term network of multiple FTIR systems (i.e. performance, stability, reliability and logistics). Such information is not in any current peer reviewed research papers, technical manuals or the instrument manual, thus whilst detailed we are not repeating material (and hopefully seen as a welcome addition to FTIR analyser literature). This manuscript could be viewed as a continuation of FTIR analyser performance reports by Griffith et al., 2012 and Hammer et al., 2013. Long term performance was not in the scope these papers. We also believe such detailed investigation and reporting are within the AMT journal remit and scope. AMT papers such as Andrews et. al., 2014 and Winderlich et al., 2010 are similar in aim, manuscript length, technical/detail scope and evaluation techniques.

I was unable to determine why some sections of the paper were put into appendices whereas other sections were not.

Sorry that this was not clear. Appendices A and B are related to the MALT spectral retrieval analysis, common to all FTIR systems, not just to the Lauder prototype. These retrieval improvements have not been published (in peer viewed literature) before. Appendix C describes the custom-made calibration suite. Details of such are not required in the main body. Appendix D describes the method used to determine local baseline conditions. It was easier to put into the appendices as baseline is mentioned prior to its determination, hence referencing to an appendix is more efficient and logical than detailing it on first mention (in section 2, Location). Routine maintenance details were moved to Appendix E to reduce manuscript size and increase readability, but still relevant technical details for FTIR analyser operation.

I recommend that the authors consider putting more of section 5 into appendices, and in the main body state only information that is required to understand the time series analysis.

This is a good suggestion, but the focus of the paper is not just to provide support/interpretation for time series analysis, but to also understand instrument operation (performance, stability, reliability) as whole along with practicalities of logistical servicing.

Section 5 is abnormally large compared to other manuscript sections. To reduce the size, section 5.4 (routine maintenance) has been moved to the appendices (as appendix E). Details about intel port configuration (section 5.2.3) and the front-end pump (section 5.2.4) have been reduced. Additionally, details pertaining to the air sampling line maintenance in section 4 have been moved to appendix E.

We decided sections on interferometer performance, Data QC/QA filtering and concentration dependent bias (sections 5.4, 5.10 and 5.8.3 respectively) should stay in the manuscript as such FTIR analyser details have not been published before and pertinent in assessing instrument performance (and in interpretation of the dataset time series). The detailed section on instrument upgrades (section 5.2) is needed as the changes mentioned have a large impact on the dataset time series uncertainty estimates and provide a heuristic link between the operation of FTIR analysers prior and post cell pressure flow decoupling. There still are FTIR systems in use (worldwide) that have not been upgraded.

We would have liked to reduce the amount of detail in section 6 but like the FTIR this is the first time the Lauder flask sampling system and related time series has been published. We cannot simply reference already published material, hence the length of this section.

Changes to the manuscript:

Appendix E added.

In the last paragraph of section 5.3 we have added a sentence stating:

"Extended periods of automation are possible (such as at remote unmanned sites) with a different measurement schedule but given that the FTIR is located on-site and accessible, regular checks and intervention are not an issue. **Details on routine maintenance can be found in appendix E.**"

Section 5.2.3 and 5.2.4 has been shortened (reduction in technical detail).

Details pertaining to the air sampling line maintenance in section 4 have been moved to appendix E.

All figures in the manuscript from figure 7 onwards have been relabelled, as figures 5 and 6 are now figures E2 and E3. Sections 5.4 to 5.12 have been relabelled due to section 5.4 now appendix E.

Note: All further replies to comments are relate to the new section and figure numbering in the manuscript.

Specific comments:

P2L12: You may want to motivate your work by reminding the reader that there are few emissions in the SH, so these SH mid-latitude measurements are crucial for pinning down the true background values.

Thanks for this advice. There is an indirect referral (via the references) to this point in the introduction: "There is also a need for increased coverage in the southern hemisphere (Thompson et al., 2014; Wells et al., 2015), which is relatively data sparse compared to the northern hemisphere." and a direct referral: "Such conditions also make it an ideal site for clean air trace gas observations.". In section 2

we state: "All these conditions make Lauder an ideal site to take baseline measurements (baseline conditions are defined Appendix D)."

To strengthen and reiterate this point in the summary (section 9) the manuscript has been changed to read:

"Despite these misgivings, the current FTIR system employing single WS calibrations is sufficient to capture $CH_4$, CO and $N_2O$ seasonal and annual trends in southern hemisphere **mid-latitude baseline** atmospheric composition within GAW reproducibly guidelines."

Whilst we state Lauder is a good site for 'clean air' measurements we do not explicitly mention that southern hemisphere *(anthropogenic?)* emissions are less than the northern hemisphere.

Technical comments:

P14L17: Did you also assess the modulation efficiency of the FTIR, along with the phase and FOV?

For the fitting of the Lauder spectra the modulation efficiency is not retrieved. The FOV is fitted instead as this gives better fits due to more consistent and lower fit residuals. This does not mean the ILS is not monitored as the fitted FOV is effectively acting as a proxy ILS diagnostic. Since the input aperture is fixed (all measurements made at with the same aperture size, 1.5mm), and the focal length of the IRcube input optics is constant, the FOV is a static quantity. Fits of the FOV should be constant. The ILS is dependent on the FOV (see Griffiths, 2007), thus any changes in the fitted FOV are indicative of a probable change in the ILS, which in practice, means a change in the alignment.

The manuscript (section 5.4) has been changed to read:

"The field of view (FOV) and spectrum phase are fitted to monitor of linewidth and asymmetry. **The ILS modulation efficiency is not retrieved. The FOV is fitted instead, as this gives more consistent and lower fit residuals whilst effectively acting as an ILS diagnostic, i.e. changes in the fitted FOV are indicative of an ILS alignment, acquisition or analysis issue.** The fitted FOV and phase are displayed in Fig. **5**c. There is a gradual decline in phase, but the overall phase is very small (< 0.01 rad) indicating a stable near symmetric ILS. The small step changes in phase are related to a change in the cell temperature sensor, laser replacement and operation of the FTIR with a different FOV. The theoretical FOV of the IRcube is unvarying at 21.73 mrad, (apart from brief testing period in mid-2011). . Prior to September 2011 the calculated FOV was lower than expected but still stable. This was because the background spectra acquisition aperture setting (3 mm) differed from the sample spectra acquisition aperture setting (1.5 mm). The background aperture size was set to 1.5 mm in September 2011. After this change the fitted FOV agrees well with the physical FOV."

P15L25: I would have liked to see more of these linear regression curves. You show one in the appendix, so either refer to that figure or plot a few more here.

On P16L4, we state that only Pressure RCS corrections are used. An example of an $N_2O$ $RCS_p$ linear regression curve is given in Fig. A1b. This figure is now referred to in the manuscript.

Section 5.6.1 in the manuscript has been changed to read (this change is part of more an extensive manuscript change to answer another question by referee 1):

"**The linear regression includes errors in the measured pressure and dry mole fraction measurement spread. For example, Fig. A1b displays the retrieved $N_2O$ dry mole fraction as a function of cell**

**pressure from tests conducted in December 2013, the resulting RCS$_p$ is 0.005 ±0.0008 ppb hPa$^{-1}$ (from table 1).**"

Thanks for spotting this.

The manuscript has been changed to read:

"This is not uncommon, both H13 and Lebegue et al. (2013) also found such experiments challenging. With strict **data quality assurance and quality control (QC/QA)**, based on cell temperature and retrieved water absolute amounts along with the relative difference between sample and calibration amounts, the associated RCS corrections are minimised **(QC/QA filtering detailed in section 5.10)**."

Also altered in the manuscript the section header:

"**5.10** Data quality assurance and quality control "

Good point, we state that "Experimental determination of RCS$_p$ is easily done" but then neglect to tell the reader how we do it. Sentences have been added to the manuscript to explain how data for RCS$_p$ derivation is obtained, how RCSp is calculated along with an example.

The manuscript has been changed to read:

"Pressure RCS (RCS$_p$) corrections need to be applied as cell pressure during sample and calibration measurements differ up to 100 hPa prior to cell pressure and flow decoupling (Fig. 4a). Experimental determination of RCS$_p$ **is performed by taking repeated measurements of dry cylinder air (usually the TC or WS) at different cell pressure, at stepped pressure increments, spanning the cell pressure operational range (see table 1). Other factors such as cell flow rate and cell temperature are held as constant as possible. Multiple measurements per pressure step are taken and averaged. The RCS$_p$ is the gradient from a simple linear regression of the retrieved dry mole fraction (response) to the cell pressure (predictor). The linear regression includes errors in the measured pressure and dry mole fraction measurement spread. For example, Fig. A1b displays the retrieved N$_2$O dry mole fraction as a function of cell pressure from tests conducted in December 2013, the resulting RCS$_p$ is 0.005 ±0.0008 ppb hPa$^{-1}$ (from table 1).** "

P18L3: retrieved dry mole fractions *to* that of the assigned

Thanks for spotting this. The grammatical error in the manuscript has been fixed on L2 (not L3).

The manuscript has been changed to read:

"From the measurements of the calibration gas an instrument response function (IRF) is constructed to map the retrieved dry mole fractions **to** that of the assigned value."

Yes, most definitely. Thanks for spotting this error.

The manuscript has been changed to read:

"The FTIR has been shown to have a linear response (H13) thus the IRF can be approximated by a first-degree (linear) polynomial, as in Eq. (**2**)."

P19L28: A step change is an indication *of* an acute incident

Thanks for spotting this. The grammatical error in the manuscript has been fixed.

The manuscript has been changed to read:

"A step change is an indication **of** an acute incident in the FTIR, FTIR acquisition procedure or a WS change"

P22L27: Suggestion: "Our approach is to take regular measurements..."

Thanks for the grammatical suggestion, it reads a lot better (eliminates the double use of the word 'take').

The manuscript has been changed to read:

"**Our approach is to take regular measurements** of a target cylinder."

P26L16: This approach would *be* need*ed* when comparing...

Thanks for the grammatical suggestion.

The manuscript has been changed to read:

"This approach would **be needed** when comparing the measurements taken in conditions of high variability (i.e. during nocturnal boundary layer inversion events)."

Figure captions: Please make the figure captions self-explanatory. For example, Fig 10 shows scaling factors, but does not discuss what is being scaled.

We also took the referee's advice to review all figure captions (taking into account information provided in figure legends) to provide better clarity and consistency.

For the example given in figure 8, the term calibration is added in front of 'scale factor'. Also, the abbreviation 'SF' is not mentioned/referenced in the manuscript. 'SF' has been replaced with '$A_{sf}$', which is referenced in the manuscript (section 5.8). In the figure 8 a, b legends, 'SF' is also replaced with '$A_{sf}$'. A new figure 8 has been inserted into the manuscript.

The caption for Figure 8 has been changed to read:

[revised manuscript text omitted]

Lastly, nine figures contain grey vertical dashed lines corresponding to instrument events. Only in the first instance (fig. 3) is an explanation given in the figure caption. The subsequent eight figure captions omit a description. We decided that repetition was not needed, as it explained in the first instance and explained in the manuscript (section 5.11). Upon the editors' decision, a repetitive descriptor per figure can be added.

Fig 17: I had trouble seeing the blue triangles. Could you make them bigger?

Sure. The blue triangle (flask data) symbol size is increased by 33% in Figs. 15 a, b, c (any larger they could be disproportionate). The new figures are incorporated into the revised manuscript.

As an example, the new Fig 15a is displayed directly below, and the old fig15a beneath it (for comparison).

Fig15. (A) new:

[Figure]

Fig15. (A) old:

[Figure]

References mentioned:

Andrews, A. E., Kofler, J. D., Trudeau, M. E., Williams, J. C., Neff, D. H., Masarie, K. A., Chao, D. Y., Kitzis, D. R., Novelli, P. C., Zhao, C. L., Dlugokencky, E. J., Lang, P. M., Crotwell, M. J., Fischer, M. L., Parker, M. J., Lee, J. T., Baumann, D. D., Desai, A. R., Stanier, C. O., De Wekker, S. F. J., Wolfe, D. E., Munger, J. W., and Tans, P. P.: $CO_2$, CO, and $CH_4$ measurements from tall towers in the NOAA Earth System Research Laboratory's Global Greenhouse Gas Reference Network: instrumentation, uncertainty analysis, and recommendations for future high-accuracy greenhouse gas monitoring efforts, Atmos. Meas. Tech., 7, 647-687, https://doi.org/10.5194/amt-7-647-2014, 2014.

Griffith, D. W. T., N. Deutscher, C. Caldow, G. Kettlewell, M. Riggenbach, S. Hammer, A Fourier transform infrared trace gas analyser for atmospheric applications, Atmos. Meas. Tech., 5, 2481–2498, 2012

Hammer, S., D. W. T. Griffith, G. Konrad, S. Vardag, C. Caldow, I. Levin, Assessment of a multi-species in situ FTIR for precise atmospheric greenhouse gas observations, Atmos. Meas. Tech., 6, 1153–1170, 2013

Winderlich, J., Chen, H., Gerbig, C., Seifert, T., Kolle, O., Lavrič, J. V., Kaiser, C., Höfer, A., and Heimann, M.: Continuous low-maintenance $CO_2/CH_4/H_2O$ measurements at the Zotino Tall Tower Observatory (ZOTTO) in Central Siberia, Atmos. Meas. Tech., 3, 1113-1128, https://doi.org/10.5194/amt-3-1113-2010, 2010.

---

## Author Comment (AC2) · 13 Dec 2018

Reply to anonymous referee #2

We appreciate the time taken for anonymous referee #2 to carefully read and evaluate our manuscript. We thank her/him for the helpful comments and suggestions to clarify issues and to improve the content, readability and presentation of the manuscript. Below we address each question, suggestion, correction or criticism individually. Referees' comments are shown in blue. Responses are in regular font. Quotes from the manuscript are in quotation marks, with altered manuscript wording given in bold type. References referred to in replies are listed at the end.

Smale et al. describe and document in this paper a ten year time-series of continuous Greenhouse Gas mole fractions measured using a FTIR analyser at Lauder, New Zealand. They describe the improvements introduced to the measurement setup and the instrument and evaluate how these affected the measurement precision and accuracy. Unfortunately they do not describe the results for CO2 and 13CO2 in this paper and focus only on CH4, CO and N2O.

We agree, it would have been nice to include $CO_2$ and $\delta13C-CO_2$, but these datasets are not ready for publication. We have not concluded $CO_2$ error characterization (details can be found in Smale et. al., GAW report 206, 2012 and Smale et. al., GAW report 213, 2014). The current $\delta13C-CO_2$ spectral retrieval analysis and calibration strategies are currently not fit for publication. A complete reanalysis is required using analysis and calibration methodologies prescribed by Griffith, 2018. We intend to do this.

General comments:

I concur with reviewer #1 that the paper is generally well written but way too long. Many of the detailed descriptions could be abbreviated with at least 50% or be transferred into the appendices (e.g. sections 5.4 and 5.5).

Reduction of manuscript length was also recommended by referee 1. Section 5.4 has been moved to the appendices (as Appendix E). We feel that section 5.5 should remain in the manuscript as opposed to the appendices as no previous (UoW/Spectronus FTIR analyser) published study has investigated interferometer performance. We hope that groups using the FTIR analyser/Spectronus will start to routinely look at interferometer parameters as part of overall QC/QA diagnostics.

The overall aim of the manuscript is a detailed investigation of FTIR long term performance. We feel a 50% reduction in content would seriously detract from the amount of detail needed to meet the aim of this work.

We have additional responses to this comment, which are the same as given to referee 1. To save repetition please see replies to referee 1 comments on the same topic (i.e. manuscript reduction).

Changes to the manuscript (same as in reply to referee 1):

Appendix E added.

In the last paragraph of section 5.3 we have added a sentence stating:

"Extended periods of automation are possible (such as at remote unmanned sites) with a different measurement schedule but given that the FTIR is located on-site and accessible, regular checks and intervention are not an issue. **Details on routine maintenance can be found in appendix E.**"

Section 5.2.3 and 5.2.4 has been shortened (reduction in technical detail).

Details pertaining to the air sampling line maintenance in section 4 have been moved to appendix E.

All figures in the manuscript from figure 7 onwards have been relabelled, as figures 5 and 6 are now figures E2 and E3. Sections 5.4 to 5.12 have been relabelled due to section 5.4 now appendix E.

Note: All further replies to comments are relate to the new section and figure numbering in the manuscript.

Although the paper claims that this is the longest time series from this kind of instrument to date, this could be taken with a grain of salt, one could argue that the actual homogeneous time series only starts after the many changes in setup that took place up until Feb 2014.

We thank the referee for culinary advice on how to season our time series, but we disagree with this comment as there are numerous high precision in situ (and remote sensing) time series that span multiple decades comprised of measurements taken with multiple successive instruments but, as a whole, provide single long-term datasets. For example, datasets in Brailsford et al., 2012, Liley et al., 2000 and Prinn et al., 2000

Common to all instruments, parts will be replaced and upgraded to improve performance. In this study each $CH_4$, CO and $N_2O$ dataset is essentially homogeneous as across the respective time series the same spectral analysis and calibration methodologies have been employed, a common static RCSp sensitivity is used and all working tanks and target cylinders are on the same scale. Instrument upgrades have improved accuracy and precision, but we do not think this disqualifies it from being considered a single time series.

Stability greatly improved after cell pressure could be actively held at 1100 hPa (Sept 2013), but there have been continual small improvements to the system before and after this major change. How do we define a change that make time series inhomogeneous? Each individual WT change will also introduce a small systematic bias (the largest factor in systematic uncertainty is the WT assignment uncertainty). Is this a discontinuity in homogeneity?

We think we are justified that the statement "Being the longest continuous deployed operational FTIR system…" is not an exaggerated or false claim.

However, the careful evaluation of measurement biases and precision as a function of time as performed here are a significant improvement over just providing the mole fraction time-series, and should be recommended good practice for all published GAW in-situ observations.

Specific comments:

The paper refers in the abstract to the compatibility goals as set by WMO GAW for greenhouse gas observations and compares the most recent results after all improvements and fine tuning to these by looking at the comparison with analyses of flask samples. Although the comparison with flask samples is a useful and common measure for quality assurance it is not the most authoritative measure.

We agree, flask sampling is not the most authoritative in situ measurement. We state reasons why we choose to start a parallel flask sampling: "Routine (weekly) in situ flask air sample collection at Lauder started in May 2009 as a robust proven cost-effective approach to provide independent measurements of $CH_4$, CO, $N_2O$, $CO_2$ and $\delta^{13}C$-$CO_2$ for comparison against FTIR measurements." and additionally state the drawbacks of using flask samples: "One drawback of flask sampling is that measurements are not continuous, offering only a sparse temporal dataset."

As in section 6 the rejection criterium for the duplo analysis of the flasks for N2O at the NIWA GC system is set to 0.5 ppb, I assume that the GC analysis reproducibility is about half of this and thus we cannot expect that the comparison between flask and FTIR measurements will be conclusive and be better than this 0.3 ppb. A better comparison would be to compare with in-situ continuous observations with different techniques such as CRDS or QCL or results of one of the round-robin exercises from the GAW CCL.

An error in the manuscript was spotted in the reporting of the GC $N_2O$ duplo flask rejection criteria of 0.5ppb, it should be 0.4ppb. This does not alter the reported analysis or results as the error was only in the manuscript.

The manuscript (section 6) has been changed to read:

"Samples with intra-flask differences greater than the combined uncertainty in each sample pair are rejected or if flask difference exceed 2.0 ppb, 1.0 ppb and **0.4** ppb for $CH_4$, CO and $N_2O$ respectively."

This quantitative change does not alter the referee's valid point that comparison results must consider the combined measurement uncertainties. The so called authoritative dataset (flask) has uncertainties. Comparison of the measurement differences to that of the GAW recommended compatibility goals also must take into consideration the FTIR measurement uncertainties along with the authoritative dataset uncertainties. The combined FTIR flask difference uncertainty (illustrated as error bars in fig 14 a, d, g) are calculated using both FTIR and flask measurement uncertainties and sample period variability. The uncertainty in the $CH_4$ FTIR flask measurement differences are comparable in magnitude to the GAW recommend compatibility goal and for CO, the uncertainties are less. For $N_2O$, the FTIR flask measurement difference uncertainties are greater than the GAW recommended compatibility goal of 0.1ppb. Achieving this goal may be unobtainable given the current FTIR and flask sampling $N_2O$ systematic and random uncertainty components. We agree that a comparison of FTIR measurements against that of another high precision in situ continuous system at Lauder would be very beneficial, especially for $N_2O$

We have added to the manuscript (section 7) these points:

"For $N_2O$, a bias of -0.01 ± 0.77 ppb is within the GAW recommended compatibility goal of 0.1ppb but this is more serendipitous when the FTIR flask time series and correlation scatter plots are viewed (Fig. **14** g, h). **Any comparison of bias to that of the GAW recommended compatibility goal also must take into consideration the FTIR and flask measurement uncertainties. In each $N_2O$ FTIR flask comparison, the uncertainties (error bars in fig 14 a, d, g) are greater than the GAW recommended compatibility goal of 0.1ppb. Achieving combined uncertainty estimates less that the compatibility goal may be unobtainable given the current FTIR and flask sampling $N_2O$ systematic and random uncertainty components**. Care must **also** be taken in interpretation as systematic differences dominate in different time periods, but as an ensemble, produce statistical results that could convey a large, but Gaussian spread (Fig. **14**i). For instance, there is an increased bias over the time interval 2014.65-2016.08. So far, the causes are unknown. There is no explicit correlation between the bias with any FTIR instrument or flask sample events, and only affects $N_2O$ (not CO or $CH_4$). We suspect the issue is with the FTIR measurement as the elevated level of $N_2O$ is greater than what simple trend analysis would indicate, as seen in the baseline time series (see Fig. **15**c). There is also a sudden (step) decrease of $N_2O$ at the start of 2016 that is not seen in the $N_2O$ flask samples.

$N_2O$ FTIR comparison measurements carried out by Griffith et al., 2011 show much better results. A bias of -0.12 ppb was also reported but with a standard deviation of 0.22 ppb. $N_2O$ FTIR comparisons conducted by Vardag et al. (2014), also report a much smaller standard deviation (0.22 ppb) than our

results. A comprehensive investigation of five continuous $N_2O$ analysers (including the FTIR) by Lebegue et al. (2016), showed FTIR performance comparable to the other instruments. These findings point to a specific but as yet unidentified issue with the Lauder FTIR $N_2O$ measurements.  Internal FTIR QC/QA did not identify any issues over the 2014.65-2016.08 period. **Overall, for $N_2O$, such independent validation via flask sampling comparisons may not be of sufficiently low uncertainty or high enough temporal resolution to address issues. Comparisons at a greater temporal resolution, such as another high precision in-situ continuous system operating in parallel, may assist in resolving disparities encountered and reduce combined uncertainty estimates.**"

And in section 9 (conclusion):

"Comparison of FTIR and co-located flask measurements show good agreement for $CH_4$ and CO. Whilst the bias of $N_2O$ FTIR flask comparisons is within GAW recommended compatibility goals, this is serendipitous and dominated by systematic differences. **A comparison campaign at Lauder using another high precision continuous $N_2O$ in situ instrument would be advantageous.** Simplistic baseline time series trend analysis was conducted with calculation of linear annual trends and seasonal cycles. The deduced trends and seasonal cycles align with estimates from other southern hemisphere in situ measurements."

Lastly, being part of GAW CCL round robin as already been proven to be beneficial. NIWA Gaslab is part of such activities, which highlighted issues in $N_2O$ working tank assignments: "A 0.65ppb bias was observed in WCC-$N_2O$ travelling standard measurements at NIWA-Gaslab during an audit of the Baring Head GAW station in 2009 (Scheel, 2012).". FTIR measurements of such round robin tanks would highlight this issue locally at Lauder. The ANIWANIWA tank suite performs a quasi-round robin role as suite assignment was done at NOAA GMDL, independent of NIWA Gaslab.

Technical comments:

P8L8: for the PT100 RTD one should specify the tolerance class, the resolution of the transmitter is not that relevant as long as it is order of magnitude better than the tolerance class value. From the value specified in P8L16 one might guess the tolerance class is F 0.1.

The PT100 RTD (flat film) has a 'Class A' tolerance value. The tolerance nomenclature was revised in the IEC 60751 2008-0 international standard, thus the new tolerance designation of 'Class A' is 'F0.15'.

The acronym 'RTD' is also removed as it is not required further on in the manuscript.

Section 5.1 in the manuscript has been changed to read:

"The FTIR enclosure is thermostatically controlled, with a manual set point at 34.0 °C. Cell temperature was originally monitored with a LM335 integrated circuit sensor attached to the outside of the cell (resolution 0.1 °C) later replaced with  **more precise** in-cell **temperature** sensor**s** as described further below."

Along with changes in Section 5.2.1

"In September 2010, a PT100 **(tolerance class F0.15)** resistance thermometer detector **()** was inserted into the cell to measure gas temperature invitro."

P8L19: A thermocouple will show significant more short term and long term drift than any PT100 so the reason for this change is questionable. There also very thin, fast response time, PT100 RTDs.

The original reason for the replacement was to use a temperature probe with a faster response to allow investigation of temperature disequilibrium effects and to bring the Lauder FTIR prototype componentry more in line with the Spectronus FTIR system (which uses the Type-J thermocouple). The biggest 'step' in temperature monitoring was the replacement of the external LM335 with faster response in vitro probes.

In hindsight, we agree, a change from the PT100 to the thermocouple was not needed as both sensors gave similar readings during temperature disequilibrium testing, and in standard operating conditions both sensors give similar readings (see fig 3., before and after April 2013). This is stated as such in Section 5.2.1: "Even though the thermocouple has a faster response time, no significant changes in temperature precision were seen."

Any sensor drift is undesirable. Small long-term temperature drift will not affect the calibrated timeseries, as the calibration method will effectively cancel any drift (assuming the drift effects calibration and sample measurements in the same manner). Given the dataset used in this research we cannot explicitly diagnose any long-term (or short term) temperature drift. The 10-minute averaged cell temperature from 2014.0 to 2017.0 is displayed in the figure below. The dataset is split into two at ~2015.3. This is when there was a substantial change in the laboratory air conditioning which effected the FTIR enclosure temperature, hence cell temperature. The red and blue subsets are cell temperatures with 6-sigma outliers removed pre and post laboratory temperature change. The green and orange lines are linear fits to the red and blue subsets respectively. The linear temperature trend prior to 2015.3 was ~-0.006 °C year$^{-1}$ and 0.007 °C year$^{-1}$ post 2015.3. The trend cannot be completely attributed to sensor drift, as cell temperature maybe slowly varying, but the current analysis is a good indicator of upper limits on temperature sensor drift, and if so due to the small magnitude then such drift will be easily compensated for in the calibration method.

[Figure]

Fig 1. Cell temperature (10-minute average) over three years (2014.0-2017.0). The data set is split into two at ~2015.3. This is when there was a substantial change in the laboratory air temperature which effected the FTIR enclosure temperature, hence cell temperature. The red and blue subsets are cell temperatures with 6-sigma outliers removed pre and post laboratory temperature change. The green and orange lines are linear fits to the red and blue subsets respectively.

P10L35:P11L8: There will be a small residual of sample air (1/200*1/870) left in the WS and TC air samples, is this corrected for in the analyses by using the mole fractions determined in the previous sample?

We do not correct for prior sample residual in WS and TC tank measurements. The main reason is that (assuming complete mixing) the proportion of prior sample residual (psr) is very small and the concentration difference between consecutive measurements is small (relative to the psr). See the equation below.

$C_{t\_corr} = C_{t-1}*psr + C_t*(1-psr) = C_t + psr*(C_{t-1}-C_t)$

where psr = proportion of prior sample residual

$C_{t-1}$ = species concentration of prior measurement

$C_t$ = species concentration of present measurement

$C_{t\_corr}$ = corrected current measurement

In all cases $(C_{t-1}-C_t) << (1/psr)$ so $C_{t\_corr} \sim= C_t$. For example, with a psr $\sim= 1/200 * 1/870 \sim= 1/174000$ (worst case), or 1/220000 at 1100 hPa (post April 2013) and an overly exaggerated $(C_{t-1}-C_t)$ of 1000ppb (rare occurrence but possible for $CH_4$ between a WS or TC and a sample taken during nocturnal boundary conditions) the correction would be 1000/174000 $\sim=0.006$ppb.

Any applied prior sample concentration correction is well below the FTIR accuracy and precision limits (and respective systematic and random uncertainty estimates). In appendix A of Hammer et al. (2013) prior sample residual proportion (called sample memory effect) of ~0.02% was calculated and not corrected for.

The manuscript (section 5.3) has been changed to read:

"Prior to WS tank measurement the cell is flushed with 200 hPa of WS gas then the cell is re-evacuated to 1 hPa and filled to the prescribed pressure set point. **In this double stage evacuation, the prior sample memory effect is less than 0.001%.** Filling takes approx. 60 seconds."

P10L35:P11L8: Why were the WS and TC measurements not performed in duplo or triplet? This would allow to detect offsets due to differences between flow and static mode especially for the first filling due to for example differences in water vapor content, this was recognized by the authors as since Feb 2014 the first calibration result is always skipped (P11L24). How big was the effect there?

WS and TC measurements are performed in triplicate after a change in the calibration routine in Feb 2014. With the benefit of hindsight, we should have taken triplicate measurements prior to Feb 2014, but we did not know about the temperature disequilibrium effect and did not know the extent of any static-flow differences. In both cases numerous tests were conducted to quantify these effects which led to the standard operating procedure change in February 2014. Both the temperature disequilibrium effect and change in measurement modes had a large statistically significant effect on measured $CO_2$ (hence another reason to withhold the current $CO_2$ dataset until we do more work in it).

As mentioned in the manuscript we note no statistically significant differences in $CH_4$, CO and $N_2O$ WT measurements due to the temperature disequilibrium effect. We neglected to mention the effect of any static-flow mode measurement differences. Tests showed no statistically significant differences in CO and $N_2O$ measured in static and flow modes.  There were statistically significant differences in the measured $CH_4$ in all static-flow tests we conducted, but no consistent systematic bias across the tests. The static-flow biases, per test, ranged from ~-0.3 to 0.45 ppb. Due to the variability in the biases we cannot determine an overall systematic bias, but we can account for it as a random uncertainty.

We neglected this component in the analysis and presentation in the manuscript (we thank the referee bringing it to our attention!). Data was reprocessed with an additional $CH_4$ calibration random uncertainty of 0.5 ppb (a conservative estimate, added in quadrature with current terms) in data prior

to February 2014. This propagates directly into scale factor uncertainty (fig 8b) and the measurement uncertainty budget (fig 13a).

Manuscript changes: Figures 8b and Figure 13a were changed due to reprocessing of data.

The manuscript (section 5.3) has been changed to read:

"The combined slower fill rate and longer settling time allows cell temperature and pressure to stabilise with a significant reduction in thermodynamic disequilibrium. The effect of thermodynamic disequilibrium has minimal impact on $CH_4$, CO and $N_2O$ spectral analysis but significant for $CO_2$. **Additionally, on the change from static to flow calibrations there were no statistically significant differences in CO and $N_2O$ WT measurements. There were statistically significant differences in $CH_4$ WT measurements. Tests conducted showed static-flow biases ranging from -0.3ppb to 0.45ppb. The reasons for spread in the bias are unknown. We have included an additional random uncertainty term of 0.5 ppb prior to Feb 2014 in the $CH_4$ WT uncertainty budget calculation to account for the fact measurements were taken in flow mode whilst calibrations were conducted in static mode.**

Once the cell is filled, tank gas flows at a rate of 0.5 $Lmin^{-1}$ during which spectra measurements are taken. Four 10-minute spectra are collected. The first is not used, effectively allowing another 10 minutes for the FTIR to stabilise"

The manuscript (section 5.8.2) has been changed to read:

"There is an increase in the $CH_4$ scale factor **variability** after 2014. This has been attributed to an error in the background spectrum $H_2O$ stripping procedure. This affects both sample and calibration measurements equally hence the calibrated sample measurements remain unaffected. **Conversely, there was a reduction in $CH_4$ scale factor uncertainty variability after 2014 due to changes in standard operating conditions**. Longer term gradual scale factor changes are harder to diagnose. The **reason for the** gradual decline in the $CH_4$ and $N_2O$ scale factors from 2007 to 2010 is unclear. Hypothesis include MIR globar intensity deterioration, cell wall effects and pressure/temperature sensor drift. The decline spans multiple WSs and instrument changes."

The manuscript (section 5.11) has been changed to read:

"Figure **13** displays the total, systematic and random uncertainties of the calibrated timeseries for each species. The average uncertainty is approx. **1.5** ppb, 0.6 ppb, and 0.3 ppb for $CH_4$, CO and $N_2O$ respectively, with uncertainty proportional with measurement concentration (due to error propagation). The short duration large spikes in uncertainty are related to instances of high sample measurement concentrations in which uncertainties propagate. For two instances in the $CH_4$ record (at the start of 2007 and 2014) the large uncertainty is due to a larger than usual scale factor uncertainty. **The reduction in $CH_4$ random uncertainty after February 2014 is due a switch from static to flow mode calibrations.** Since the upgrade in April 2013 $RCS_p$ corrections for all species have been negligible, hence a reduction in associated uncertainty. "

P19L28: indication an -> indication of an

Thanks for spotting this along with Anonymous Referee #1.

The manuscript has been changed to read:

"A step change is an indication **of** an acute incident in the FTIR, FTIR acquisition procedure or a WS change"

P19L34: approx. -> approximate

Again, thanks.

The manuscript has been changed to read:

"For example, in mid-2011 there was an **approximate** 3% increase in the $N_2O$ scale factor for a short period."

P30L25: The link given to the data will become obsolete after November 2018, as this website will be shutdown by JMA. The new WDCGG site is: https://gaw.kishou.go.jp/. It would be good to have the total uncertainty and bias estimates as in figure 15 also available together with the mole fraction time series in the same file or as a separate datafile.

A good snippet of advice, many thanks.

The manuscript has been changed to read:

"Calibrated baseline $CH_4$ FTIR and $CH_4$ flask sample measurements **are archived in the World Data Centre for Greenhouse Gases database (https://gaw.kishou.go.jp).**"

We will also endeavour to include uncertainty (total, systematic and random) estimates in future data submissions.

Figure 16 there seems to be a cluster of obs for N2O where flask measurements are higher than the FTIR. It would be useful to see if the lower ring of dots below the 1:1 line between flask 325-328 ppb and FTIR 325-327 is a cluster connected in time that could be removed due to a problem in either GC or FTIR obs.

A lot of time was spent on FTIR and GC data QC/QA. This particular issue was identified during analysis, along with possible erroneous flask outliers in late-2010. In these cases, we could not find any diagnostic or correlation with a specific instrument event pointing to erroneous data collection. The only indication was the measurand itself. It is hard to plausibility defend removal of data without a good cause (especially when it improves the bias). Whilst there maybe causes we cannot currently identify them, thus all data passing QC/QA criteria is used. All this illustrates the variability and toughness of making such long-term measurements and the current state of $N_2O$ measurements (FTIR and flask) at Lauder.

In the case of the $N_2O$ data in the interval 2014.65-2016.08, we did remove this subset but only for timeseries trend analysis (see Fig 15c). The subset is still part of the FTIR dataset and the effect on trend analysis by the removal of the time series was diagnosed (section 8 ):  "To check, the annual trend calculated with inclusion of the flagged erroneous data was estimated at 1.06 ppb year-1 (± 0.01) compared to 0.99ppb year-1, demonstrating that inclusion alters the trend estimate by approx. 6%.".

References mentioned:

Brailsford, G. W., Stephens, B. B., Gomez, A. J., Riedel, K., Mikaloff Fletcher, S. E., Nichol, S. E., and Manning, M. R.: Long-term continuous atmospheric $CO_2$ measurements at Baring Head, New Zealand, Atmos. Meas. Tech., 5, 3109-3117, 10.5194/amt-5-3109-2012, 2012.

GAW: Report no. 206. 16th WMO/IAEA Meeting on Carbon Dioxide, Other Greenhouse Gases and Related Tracers Measurement Techniques, Wellington. http://www.wmo.int/pages/prog/arep/gaw/documents/Draft_GAW_206_5_Nov.pdf ,2012

GAW: Report no. 213. 17th WMO/IAEA Meeting on Carbon Dioxide, Other Greenhouse Gases and Related Tracers Measurement Techniques, Beijing. https://library.wmo.int/pmb_ged/gaw_213_en.pdf, 2014

Griffith, D. W. T.: Calibration of isotopologue-specific optical trace gas analysers: a practical guide, Atmos. Meas. Tech., 11, 6189-6201, 10.5194/amt-11-6189-2018, 2018.

Hammer, S., Griffith, D. W. T., Konrad, G., Vardag, S., Caldow, C., and Levin, I.: Assessment of a multi-species in situ FTIR for precise atmospheric greenhouse gas observations, Atmos. Meas. Tech., 6, 1153-1170, 10.5194/amt-6-1153-2013, 2013.

Liley, J. B., et al. "Stratospheric NO2 variations from at Lauder, New Zealand a long time series." J. Geophys. Res 105.D9 (2000): 11-633.

Prinn, R. G., et al. (2000), A history of chemically and radiatively important gases in air deduced from ALE/GAGE/AGAGE, *J. Geophys. Res.*, 105(D14), 17751–17792, doi: 10.1029/2000JD900141.

Scheel, H.: GAW World Calibration Centre for Nitrous Oxide (WCC-$N_2O$) Report 2009 – 2011 FZK: 351 01 069, https://www.imk-ifu.kit.edu/wcc-n2o/docs/WCC-N2O_Report_2009-2011.pdf, 2012.